# SOLVE MINIMAX OPTIMIZATION BY ANDERSON ACCELERATION

**Huan He, Shifan Zhao, Yuanzhe Xi, Joyce C Ho** [*]
Department of Computer Science
Emory University
Atlanta, GA 30329, USA

**Yousef Saad** [†]
Department of Computer Science and Engineering
University of Minnesota
Minneapolis, MN 55455, USA

## ABSTRACT

Many modern machine learning algorithms such as generative adversarial networks (GANs) and adversarial training can be formulated as minimax optimization. Gradient descent ascent (GDA) is the most commonly used algorithm due to its simplicity. However, GDA can converge to non-optimal minimax points. We propose a new minimax optimization framework, `GDA-AM`, that views the GDA dynamics as a fixed-point iteration and solves it using Anderson Mixing to converge to the local minimax. It addresses the diverging issue of simultaneous GDA and accelerates the convergence of alternating GDA. We show theoretically that the algorithm can achieve global convergence for bilinear problems under mild conditions. We also empirically show that `GDA-AM` solves a variety of minimax problems and improves adversarial training on several datasets. Codes are available on Github [1].

## 1 INTRODUCTION

Minimax optimization has received a surge of interest due to its wide range of applications in modern machine learning, such as generative adversarial networks (GAN), adversarial training and multi-agent reinforcement learning (Goodfellow et al., 2014; Madry et al., 2018; Li et al., 2019). Formally, given a bivariate function $f(\boldsymbol{x}, \boldsymbol{y})$, the objective is to find a stable solution where the players cannot improve their objective, i.e., to find the Nash equilibrium of the underlying game (von Neumann & Morgenstern, 1944):

$$\arg\min_{x \in \mathcal{X}} \arg\max_{y \in \mathcal{Y}} f(\boldsymbol{x}, \boldsymbol{y}). \tag{1}$$

It is commonplace to use simple algorithms such as gradient descent ascent (GDA) to solve such problems, where both players take a gradient update simultaneously or alternatively. Despite its simplicity, GDA is known to suffer from a generic issue for minimax optimization: it may cycle around a stable point, exhibit divergent behavior, or converge very slowly since it requires very small learning rates (Gidel et al., 2019a; Mertikopoulos et al., 2019). Given the widespread usage of gradient-based methods for solving machine learning problems, first-order optimization algorithms to solve minimax problems have gained considerable popularity in the last few years. Algorithms such as optimistic Gradient Descent Ascent (OG) (Daskalakis et al., 2018; Mertikopoulos et al., 2019) and extra-gradient (EG) (Gidel et al., 2019a) can alleviate the issue of GDA for some problems. Yet, it has been shown that these methods can still diverge or cycle around a stable point (Adolphs et al.; Mazumdar et al., 2019; Parker-Holder et al., 2020). For example, these algorithms even fail to find a local minimax (the set of local minimax is a superset of local Nash (Jin et al., 2020; Wang et al., 2020)) as shown in Figure 1. This leads to the following question: *Can we design better algorithms for minimax problems?* We answer this in the affirmative, by introducing `GDA-AM`. We cast the GDA dynamics as a fixed-point iteration problem and compute the iterates effectively using an advanced nonlinear extrapolation method. We show that indeed our algorithm has theoretical and empirical guarantees across a broad range of minimax problems, including GANs.

---

[*]hhe37, szhao89, yxi26, jho31@emory.edu

[†]saad@umn.edu

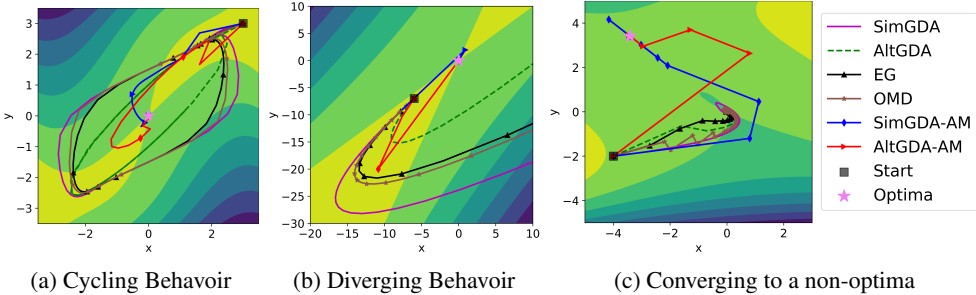

(a) Cycling Behavoir  (b) Diverging Behavoir  (c) Converging to a non-optima

Figure 1: **Left**: $f(x,y) = (4x^2 - (y - 3x + 0.05x^3)^2 - 0.1y^4)e^{-0.01(x^2+y^2)}$. **Middle**: $-3x^2 - y^2 + 4xy$. **Right**: $f(x,y) = 2x^2 + y^2 + 4xy + \frac{4}{3}y^3 - \frac{1}{4}y^4$. We can observe that baseline methods fail to converge to a local minimax, whereas GDA-AM with table size $p = 3$ always exhibits desirable behaviors.

**Our contributions:** In this paper, we propose a different approach to solve minimax optimization. Our starting point is to cast the GDA dynamics as a fixed-point iteration. We then highlight that the fixed-point iteration can be solved effectively by using advanced non-linear extrapolation methods such as Anderson Mixing (Anderson, 1965), which we name as GDA-AM. redAlthough first mentioned in Azizian et al. (2020), to our best knowledge, this is still the first work to investigate and improve the GDA dynamics by tapping into advanced fixed-point algorithms.

We demonstrate that GDA dynamics can benefit from Anderson Mixing. In particular, we study bilinear games and give a systematic analysis of GDA-AM for both simultaneous and alternating versions of GDA. We theoretically show that GDA-AM can achieve global convergence guarantees under mild conditions.

We complement our theoretical results with numerical simulations across a variety of minimax problems. We show that for some convex-concave and non-convex-concave functions, GDA-AM can converge to the optimal point with little hyper-parameter tuning whereas existing first-order methods are prone to divergence and cycling behaviors.

We also provide empirical results for GAN training across two different datasets, CIFAR10 and CelebA. Given the limited computational overhead of our method, the results suggest that an extrapolation add-on to GDA can lead to significant performance gains. Moreover, the convergence behavior across a variety of problems and the ease-of-use demonstrate the potential of GDA-AM to become the minimax optimization workhorse.

## 2 PRELIMINARIES AND BACKGROUND

### 2.1 MINIMAX OPTIMIZATION

**Definition 1.** *Point $(\mathbf{x}^*, \mathbf{y}^*)$ is a **local Nash equilibrium** of $f$ if there exists $\delta > 0$ such that for any $(\mathbf{x}, \mathbf{y})$ satisfying $\|\mathbf{x} - \mathbf{x}^*\| \leq \delta$ and $\|\mathbf{y} - \mathbf{y}^*\| \leq \delta$ we have: $f(\mathbf{x}^*, \mathbf{y}) \leq f(\mathbf{x}^*, \mathbf{y}^*) \leq f(\mathbf{x}, \mathbf{y}^*)$.*

To find the Nash equilibria, common algorithms including GDA, EG and OG, can be formulated as follows. For the two variants of GDA, simultaneous GDA (SimGDA) and alternating GDA (AltGDA), the updates have the following forms:

$$
\begin{aligned}
\text{Simultaneous}: \quad & \mathbf{x}_{t+1} = \mathbf{x}_t - \eta\nabla_{\mathbf{x}}f(\mathbf{x}_t, \mathbf{y}_t), \quad \mathbf{y}_{t+1} = \mathbf{y}_t + \eta\nabla_{\mathbf{y}}f(\mathbf{x}_t, \mathbf{y}_t) \\
\text{Alternating}: \quad & \mathbf{x}_{t+1} = \mathbf{x}_t - \eta\nabla_{\mathbf{x}}f(\mathbf{x}_t, \mathbf{y}_t), \quad \mathbf{y}_{t+1} = \mathbf{y}_t + \eta\nabla_{\mathbf{y}}f(\mathbf{x}_{t+1}, \mathbf{y}_t).
\end{aligned}
\tag{2}
$$

The EG update has the following form:

$$
\begin{aligned}
\mathbf{x}_{t+\frac{1}{2}} &= \mathbf{x}_t - \eta\nabla_{\mathbf{x}}f(\mathbf{x}_t, \mathbf{y}_t), & \mathbf{y}_{t+\frac{1}{2}} &= \mathbf{y}_t + \eta\nabla_{\mathbf{y}}f(\mathbf{x}_t, \mathbf{y}_t) \\
\mathbf{x}_{t+1} &= \mathbf{x}_t - \eta\nabla_{\mathbf{x}}f(\mathbf{x}_{t+\frac{1}{2}}, \mathbf{y}_{t+\frac{1}{2}}), & \mathbf{y}_{t+1} &= \mathbf{y}_t + \eta\nabla_{\mathbf{y}}f(\mathbf{x}_{t+\frac{1}{2}}, \mathbf{y}_{t+\frac{1}{2}}).
\end{aligned}
\tag{3}
$$

The OG update has the following form:

$$\mathbf{x}_{t+1} = \mathbf{x}_t - \eta \nabla_{\mathbf{x}} f(\mathbf{x}_t, \mathbf{y}_t) + \frac{\eta}{2} \nabla_{\mathbf{x}} f(\mathbf{x}_{t-1}, \mathbf{y}_{t-1}), \mathbf{y}_{t+1} = \mathbf{y}_t + \eta \nabla_{\mathbf{y}} f(\mathbf{x}_t, \mathbf{y}_t) - \frac{\eta}{2} \nabla_{\mathbf{y}} f(\mathbf{x}_{t-1}, \mathbf{y}_{t-1}).$$
(4)

## 2.2 FIXED-POINT ITERATION AND ANDERSON MIXING (AM)

**Definition 2.** $\mathbf{w}^\star$ *is a fixed point of the mapping $g$ if $\mathbf{w}^\star = g\left(\mathbf{w}^\star\right)$.*

Consider the simple fixed-point iteration $w_{t+1} = g(w_t)$ which produces a sequence of iterates $\{w_0, w_1, \cdots, w_N\}$. In most cases, this converges to the fixed-point, $w^* = g(w^*)$. Take gradient descent as an example, it can be viewed as iteratively applying the operation: $w_{t+1} = g(w_t) \triangleq w_t - \alpha_t \nabla f(w_t)$, where the limit is the fixed-point $\mathbf{w}^\star = g\left(\mathbf{w}^\star\right)$ (i.e. $\nabla f(w_t = 0)$. SimGDA updates can be defined as the repeated application of a nonlinear operator:

$$\mathbf{w}_{t+1} = G_\eta^{(\mathrm{sim})}(\mathbf{w}_t) \triangleq \mathbf{w}_t - \eta V(\mathbf{w}_t) \text{ with } \mathbf{w} = \begin{bmatrix} \mathbf{x} \\ \mathbf{y} \end{bmatrix}, V(\mathbf{w}) = \begin{bmatrix} \nabla_{\mathbf{x}} f(\mathbf{x}, \mathbf{y}) \\ -\nabla_{\mathbf{y}} f(\mathbf{x}, \mathbf{y}) \end{bmatrix}$$

Similarly, we can write AltGDA updates as $\mathbf{w}_{t+1} = G_\eta^{(\mathrm{alt})}(\mathbf{w}_t)$. An issue with fixed-point iteration is that it does not always converge, and even in the cases where it does converge, it might do so very slowly. GDA is one example that it could result in the possibility of the operator converging to a limit cycle instead of a single point for the GDA dynamic. A way of dealing with these problems is to use acceleration methods, which can potentially speed up the convergence process and in some cases even decrease the likelihood for divergence.

There are many different acceleration methods, but we will put our focus on an algorithm which we refer to as Anderson Mixing (or Anderson Acceleration). In short, Anderson Mixing (AM) shares the same idea as Nesterov's acceleration. Given a fixed-point iteration $w_t = g(w_{t-1})$, Anderson Mixing argues that a good approximation to the final solution $w^*$ can be obtained as a linear combination of the previous $p$ iterates $w_{t+1} = \sum_{i=0}^{p} \beta_i g(w_{t-p_t+i})$. Since obtaining the proper coefficients $\beta_i$ is a nonlinear procedure, Anderson Mixing is also known as a nonlinear extrapolation method. The general form of Anderson Mixing is shown in Algorithm 1. For efficiency, we prefer a 'restarted' version with a small table size $p$ that cleans up the table $F$ every $p$ iterations because it avoids solving a linear system of increasing size.

---

**Algorithm 1:** Anderson Mixing Prototype (truncated version)

**Input:** Initial point $w_0$, Anderson restart dimension $p$, fixed-point mapping $g : \mathbf{R}^n \to \mathbf{R}^n$.
**Output:** $w_{t+1}$
**for** $t = 0, 1, \ldots$ **do**
    Set $p_t = \min\{t, p\}$.
    Set $F_t = [f_{t-p_t}, \ldots, f_t]$, where $f_i = g(w_i) - w_i$ for each $i \in [t - p_t .. t]$.
    Determine weights $\beta = (\beta_0, \ldots, \beta_{p_t})^T$ that solves $\min_\beta \|F_t \beta\|_2$, s. t. $\sum_{i=0}^{p_t} \beta_i = 1$.
    Set $w_{t+1} = \sum_{i=0}^{p_t} \beta_i g(w_{t-p_t+i})$.
**end**

---

## 2.3 AM AND GENERALIZED MINIMAL RESIDUAL (GMRES)

Developed by Saad & Schultz (1986), Generalized Minimal Residual method (GMRES) is a Krylov subspace method for solving linear system equations. The method approximates the solution by the vector in a Krylov subspace with minimal residual, which is described below.

**Definition 3.** *Assume we have the linear system of equations $\mathbf{x} = \mathbf{b}$ with $\in \mathbb{R}^{n \times n}, \mathbf{b} \in \mathbb{R}^n$ and an initial guess $\mathbf{x}_0$. Then we denote the initial residual by $\mathbf{r}_0 = \mathbf{b} - \mathbf{x}_0$ and define the $t$th Krylov subspace as $\mathcal{K}_t = span\{\mathbf{r}_0, \mathbf{r}_0, \cdots, {}^{t-1}\mathbf{r}_0\}$.*

The $t$th iterate $\mathbf{x}_t$ of GMRES minimizes the norm of the residual $\mathbf{r}_t = \mathbf{b} - \mathbf{x}_t$ in $\mathcal{K}_t$, that is, $\mathbf{x}_t$ solves

$$\min_{\mathbf{x}_t \in \mathbf{x}_0 + \mathcal{K}_t} \|\mathbf{b} - \mathbf{x}_t\|_2 .$$

The following formulation is equivalent to GMRES minimization problem and more convenient for implementation. It computes $\widehat{\mathbf{x}}_t$ such that

$$\widehat{\mathbf{x}}_t = \arg\min_{\widehat{\mathbf{x}}_t \in \mathcal{K}_t} \|\mathbf{b} - (\mathbf{x}_0 + \widehat{\mathbf{x}}_t)\|_2 = \arg\min_{\widehat{\mathbf{x}}_t \in \mathcal{K}_t} \|\mathbf{r}_0 - \widehat{\mathbf{x}}_t\|_2.$$

Using a larger Krylov dimension will improve the convergence of the method, but will require more memory. For this reason, a smaller Krylov subspace dimension $t$ and 'restarted' versions of the method are used in practice Saad (2003).

The convergence of GMRES can be studied through the magnitude of the residual polynomial.

**Theorem 2.1** (Lemma 6.31 of Saad (2003)). *Let $\widehat{\mathbf{x}}_t$ be the approximate solution obtained at the t-th iteration of GMRES being applied to solve $\mathbf{x} = \mathbf{b}$, and denote the residual as $\mathbf{r}_t = \mathbf{b} - \widehat{\mathbf{x}}_t$. Then, $\mathbf{r}_t$ is of the form*

$$\mathbf{r}_t = f_t()\mathbf{r}_0, \tag{5}$$

*where*

$$\|\mathbf{r}_t\|_2 = \|f_t()\mathbf{r}_0\|_2 = \min_{f_t \in \mathcal{P}_t} \|f_t()\mathbf{r}_0\|_2, \tag{6}$$

*where $\mathcal{P}_p$ is the family of polynomials with degree $p$ such that $f_p(0) = 1, \forall f_p \in \mathcal{P}_p$, which are usually called residual polynomials.*

Although GMRES is applied to a system of linear equations not a fixed-point problem, there is a strong connection between Anderson Mixing and GMRES. In AM we are looking for a fixed-point $\mathbf{x}$ such that $\mathbf{G}\mathbf{x} - \mathbf{b} - \mathbf{x} = 0$ and by rearranging this equation we get

$$\mathbf{b} + (\mathbf{G} - \mathbf{I})\mathbf{x} = 0 \Leftrightarrow (\mathbf{I} - \mathbf{G})\mathbf{x} = \mathbf{b}.$$

Theorem 2.2 shows that if GMRES is applied to the system $(\mathbf{I} - \mathbf{G})\mathbf{x} = \mathbf{b}$ and AM is applied to $g(\mathbf{x}) = \mathbf{G}\mathbf{x} + \mathbf{b}$ with the same initial guess and $\mathbf{I} - \mathbf{G}$ is non-singular, then these are equivalent in the sense that the iterates of each algorithm can be obtained directly from the iterates of the other algorithm.

**Theorem 2.2** (Equivalence between AM with restart and GMRES (Walker & Ni, 2011a)). *Consider the fixed point iteration $\mathbf{x} = g(\mathbf{x})$ where $g(\mathbf{x}) = \mathbf{G}\mathbf{x} + \mathbf{b}$ for $\mathbf{G} \in \mathbb{R}^{n \times n}$ and $\mathbf{b} \in \mathbb{R}^n$. If $\mathbf{I} - \mathbf{G}$ is non-singular, Algorithm 1 produces exactly the same iterates as GMRES being applied to solve $(\mathbf{I} - \mathbf{G})\mathbf{x} = \mathbf{b}$ when both algorithms start with the same initial guess.*

Theorem 2.2 can also be generalized to the restart version of AM an GMRES as well.

# 3 GDA-AM : GDA WITH ANDERSON MIXING

We propose a novel minimax optimizer, called GDA-AM, that is inspired by recent advances in parameter (or weight) averaging (Wu et al., 2020; Yazici et al., 2019). We argue that a nonlinear adaptive average (combination) is a more appropriate choice for minimax optimization.

## 3.1 GDA WITH NAÏVE ANDERSON MIXING

We propose to exploit the dynamic information present in the GDA iterates to "smartly" combine the past iterates. This is in contrast to the classical averaging methods (moving averaging and exponential moving averaging) (Yang et al., 2019) that "blindly" combine past iterates. A naïve adoption of Anderson Mixing using the past $p$ GDA iterates for both simGDA and altGDA has the following form:

$$\text{Anderson mixing}: \quad \mathbf{x}_{t+1} = \sum_{i=0}^{p} \beta_i \mathbf{x}_{t-p+i}, \mathbf{y}_{t+1} = \sum_{i=0}^{p} \beta_i \mathbf{y}_{t-p+i}.$$

Since Zhang et al. (2021); Gidel et al. (2019b) show the AltGDA is superior to SimGDA in many aspects, we briefly summarized both Simultaneous and Alternating GDA-AM in Algorithms 2 and 3 with the truncated Anderson Mixing Algorithm 1 using a table size $p$.

| **Algorithm 2:** Simultaneous `GDA-AM` | **Algorithm 3:** Alternating `GDA-AM` |
|---|---|
| **Input:** $\mathbf{x}_0, \mathbf{y}_0$, stepsize $\eta$, Anderson table size $p$ | **Input:** $\mathbf{x}_0, \mathbf{y}_0$, stepsize $\eta$, Anderson table size $p$ |
| **Output:** $\mathbf{x}_t, \mathbf{y}_t$ | **Output:** $\mathbf{x}_t, \mathbf{y}_t$ |
| Set $\mathbf{w}_0 = [\mathbf{x}_0, \mathbf{y}_0], sx = length(x_0)$ | Set $\mathbf{w}_0 = [\mathbf{x}_0, \mathbf{y}_0], sx = length(x_0)$ |
| **for** $t = 0, 1, \dots$ **do** | **for** $t = 0, 1, \dots$ **do** |
|   $\mathbf{x}_t, \mathbf{y}_t = \mathbf{w}_t[0 : sx - 1], \mathbf{w}_t[sx : end]$ |   $\mathbf{x}_t, \mathbf{y}_t = \mathbf{w}_t[0 : sx - 1], \mathbf{w}_t[sx : end]$ |
|   $\mathbf{x}_{t+1} = \mathbf{x}_t - \eta \nabla_{\mathbf{x}} f(\mathbf{x}_t, \mathbf{y}_t)$ |   $\mathbf{x}_{t+1} = \mathbf{x}_t - \eta \nabla_{\mathbf{x}} f(\mathbf{x}_t, \mathbf{y}_t)$ |
|   $\mathbf{y}_{t+1} = \mathbf{y}_t - \eta \nabla_{\mathbf{y}} f(\mathbf{x}_t, \mathbf{y}_t)$ |   $\mathbf{y}_{t+1} = \mathbf{y}_t - \eta \nabla_{\mathbf{y}} f(\mathbf{x}_{t+1}, \mathbf{y}_t)$ |
|   $\mathbf{w}_{t+1} = \begin{bmatrix} \mathbf{x}_{t+1} \\ \mathbf{y}_{t+1} \end{bmatrix}$ |   $\mathbf{w}_{t+1} = \begin{bmatrix} \mathbf{x}_{t+1} \\ \mathbf{y}_{t+1} \end{bmatrix}$ |
|   Use Anderson Mixing with table size p to extrapolate $\mathbf{w}_{t+1}$ |   Use Anderson Mixing with table size p to extrapolate $\mathbf{w}_{t+1}$ |
| **end** | **end** |
| $\mathbf{x}_t, \mathbf{y}_t = \mathbf{w}_{t+1}[0 : sx - 1], \mathbf{w}_{t+1}[sx : end]$ | $\mathbf{x}_t, \mathbf{y}_t = \mathbf{w}_{t+1}[0 : sx - 1], \mathbf{w}_{t+1}[sx : end]$ |
| **return** $\mathbf{x}_t, \mathbf{y}_t$ | **return** $\mathbf{x}_t, \mathbf{y}_t$ |

It is important to note that the Anderson Mixing form shown in Algorithm 1 is for illustrative purpose and not computationally efficient. For example, only one column of $F_t$ needs to be updated at each iteration. In addition, the solution of the least-square problem in Algorithm 1 can also be solved by a quick QR update scheme which costs $(2n + 1)p^2$ (Walker & Ni, 2011a). Thus, from Algorithms 2 and 3, we can see that the major cost of `GDA-AM` arises from solving the additional linear least squares problem compared to regular GDA at each iteration. Additional implementation details are provided in the Appendix.

## 4 CONVERGENCE RESULTS FOR `GDA-AM`

In this section, we show that both simultaneous and alternating version `GDA-AM` converge to the equilibrium for bilinear problems. First, we do not require the learning rate to be sufficiently small. Second, we explicitly provide a linear convergence rate that is faster than EG and OG. More importantly, we derive nonasymptotic rates from the spectrum analysis perspective because existing theoretical results can not help us derive a convergent rate (see C.1).

### 4.1 BILINEAR GAMES

Bilinear games are often regarded as an important simple example for theoretically analyzing and understanding new algorithms and techniques for solving general minimax problems (Gidel et al., 2019a; Mertikopoulos et al., 2019; Schaefer & Anandkumar, 2019). In this section, we analyze the convergence property of simultaneous `GDA-AM` and alternating `GDA-AM` schemes on the following zero-sum bilinear games:

$$\min_{\mathbf{x} \in \mathbb{R}^n} \max_{\mathbf{y} \in \mathbb{R}^n} f(\mathbf{x}, \mathbf{y}) = \mathbf{x}^T \mathbf{A} \mathbf{y} + \mathbf{b}^T \mathbf{x} + \mathbf{c}^T \mathbf{y}, \quad \mathbf{A} \text{ is full rank.} \tag{7}$$

The Nash equilibrium to the above problem is given by $(\mathbf{x}^*, \mathbf{y}^*) = (-^{-T}\mathbf{c}, -^{-1}\mathbf{b})$.

We also investigate bilinear-quadratic games from a spectrum analysis perspective. In addition, we show that analysis based on the numerical range (Bollapragada et al., 2018) can be also extended to such games, although it can not help derive a convergent bound for equation 7. Detailed discussion can be found in Appendix C.1 and C.4.1.

### 4.2 SIMULTANEOUS `GDA-AM`

Suppose $\mathbf{x}_0$ and $\mathbf{y}_0$ are the initial guesses for $\mathbf{x}^*$ and $\mathbf{y}^*$, respectively. Then each iteration of simultaneous GDA can be written in the following matrix form:

$$\begin{bmatrix} \mathbf{x}_{t+1} \\ \mathbf{y}_{t+1} \end{bmatrix} = \underbrace{\begin{bmatrix} \mathbf{I} & -\eta\mathbf{A} \\ \eta\mathbf{A}^T & \mathbf{I} \end{bmatrix}}_{\mathbf{G}^{(Sim)}} \underbrace{\begin{bmatrix} \mathbf{x}_t \\ \mathbf{y}_t \end{bmatrix}}_{\mathbf{w}_t^{(Sim)}} - \eta \underbrace{\begin{bmatrix} \mathbf{b} \\ \mathbf{c} \end{bmatrix}}_{\mathbf{b}^{(Sim)}}. \tag{8}$$

It has been shown that the iteration in equation 8 often cycles and fails to converge for the bilinear problem due to the poor spectrum/numerical range of the fixed point operator $\mathbf{G}^{(Sim)}$ (Gidel et al., 2019a; Azizian et al., 2020; Mokhtari et al., 2020a). Next we show that the convergence can be improved with Algorithm 2.

**Theorem 4.1.** *[Global convergence for simultaneous* GDA-AM *on bilinear problem] Denote the distance between the stationary point $\mathbf{w}^*$ and current iterate $\mathbf{w}_{(k+1)p}$ of Algorithm 2 with table size $p$ as $N_{(k+1)p} = \|\mathbf{w}^* - \mathbf{w}_{(k+1)p}\|$. Then we have the following bound for $N_t$*

$$N_{(k+1)p}^2 \leq \rho(A) N_{kp}^2 \tag{9}$$

*where $\rho(A) = (\frac{1}{T_p(1+\frac{2}{\kappa(T)-1})})^2$. Here, $T_p$ is the Chebyshev polynomial of first kind of degree $p$ and $\frac{1}{T_p(1+\frac{2}{\kappa(T)-1})} < 1$ since $1 + \frac{2}{\kappa(T)-1} > 1$.*

It is worthy emphasizing that the convergence rate of Algorithm 2 is independent of learning rate $\eta$ while the convergence results of other methods like EG and OG depend on the learning rate.

**Remark 4.1.1.** *Both EG and OG have the following form of convergence rate (Mokhtari et al., 2020a) for bilinear problem*

$$N_{t+1}^2 \leq (1 - \frac{c}{\kappa(T)}) N_t^2,$$

*where c is a positive constant independent of the problem parameters.*

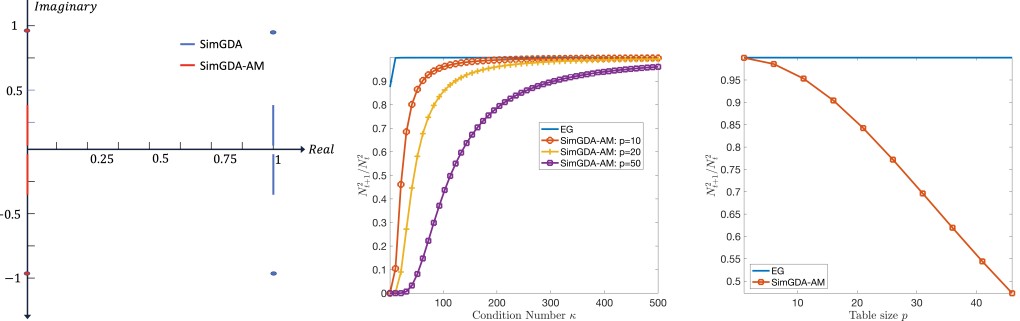

(a) Eigenvalues of iteration matrix of SimGDA and GDA-AM

(b) Different condition number

(c) Different table size, condition number $\kappa = 100$

Figure 2: **Figure 2a**: The blue line is the spectrum of matrix $\mathbf{G}^{(Sim)}$ while the red line is spectrum of matrix $\mathbf{I} - \mathbf{G}^{(Sim)}$. Our method transforms the divergent problem to a convergent problem due to the transformed spectrum. **Figure 2b**: Convergence rate comparison between SimGDA-AM and EG for different condition numbers of and fixed table size $p = 10, 20, 50$. **Figure 2c**: Convergence rate comparison between SimGDA-AM and EG for increasing table size on a matrix with condition number 100.

### 4.3 ALTERNATING GDA-AM

The underlying fixed point iteration in Algorithm 3 can be written in the following matrix form:

$$\begin{bmatrix} \mathbf{x}_{t+1} \\ \mathbf{y}_{t+1} \end{bmatrix} = \underbrace{\begin{bmatrix} \mathbf{I} & -\eta\mathbf{A} \\ \eta\mathbf{A}^T & \mathbf{I} - \eta^2\mathbf{A}^T \end{bmatrix}}_{\mathbf{G}^{(Alt)}} \underbrace{\begin{bmatrix} \mathbf{x}_t \\ \mathbf{y}_t \end{bmatrix}}_{\mathbf{w}_t^{(Alt)}} - \eta \underbrace{\begin{bmatrix} \mathbf{b} \\ \mathbf{c} \end{bmatrix}}_{\mathbf{b}^{(Alt)}}.$$

According to the equivalence between truncated Anderson acceleration and GMRES with restart, we can analyze the convergence of Algorithm 3 through the convergence analysis of applying GMRES to solve linear systems associated with $\mathbf{G} = \mathbf{I} - \mathbf{G}^{(Alt)}$:

$$\mathbf{G} = \begin{bmatrix} \mathbf{0} & \eta\mathbf{A} \\ -\eta\mathbf{A}^T & \eta^2\mathbf{A}^T \end{bmatrix}.$$

**Theorem 4.2.** *[Global convergence for alternating* `GDA-AM` *on bilinear problem] Denote the distance between the stationary point* $\mathbf{w}^*$ *and current iterate* $\mathbf{w}_{(k+1)p}$ *of Algorithm 3 with table size* $p$ *as* $N_{(k+1)p} = \|\mathbf{w}^* - \mathbf{w}_{(k+1)p}\|$. *Assume is normalized such that its largest singular value is equal to* 1. *Then when the learning rate* $\eta$ *is less than* 2, *we have the following bound for* $N_t$

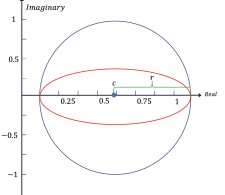

$$N_{(k+1)p}^2 \leq \sqrt{1 + \frac{2\eta}{2-\eta}} (\frac{r}{c})^p N_{kp}^2$$

*where* $c$ *and* $r$ *are the center and radius of a disk* $D(c,r)$ *which includes* all the eigenvalues of $\mathbf{G}$. Especially, $\frac{r}{c} < 1$.

Figure 3: An illustration of the spectrum of $\mathbf{G}$ (red) and the closing circle (blue) in Theorem 4.2.

Theorem 4.2 shows that when $p > \frac{\log \sqrt{\frac{2-\eta}{2+\eta}}}{\log \frac{r}{c}}$, alternating `GDA-AM` will converge globally.

## 4.4 DISCUSSION OF OBTAINED RATES

We would like to first explain on why taking Chebyshev polynomial of degree p at the point $1 + \frac{2}{\kappa-1}$. We evaluate the Chebyshev polynomial at this specific point because the reciprocal of this value gives the minimal value of infinite norm of the all polynomials of degree p defined on the interval $\tilde{I} = [\eta^2 \sigma_{min}^2(), \ \eta^2 \sigma_{max}^2()]$ based on Theorem 6.25 (page 209) (Saad, 2003). In other words, taking the function value at this point leads to the tight bound.

When comparing between existing bounds, we would like to point our our derived bounds are hard to compare directly. The numerical experiments in figure 2b numerically verify that our bound is smaller than EG. We wanted to numerically compare our rate with EG with positive momentum. However the bound of EG with positive momentum is asymptotic. Moreover, it does not specify the constants so we can not numerically compare them. We do provide empirical comparison between GDA-AM and EG with positive momentum for bilinear problems in Appendix D.1. It shows GDA-AM outperforms EG with positive momentum. Regarding alternating `GDA-AM` , we would like to note that the bound in Theorem 4.2 depends on the eigenvalue distribution of the matrix $\mathbf{G}$. Condition number is not directly related to the distribution of eigenvalues of a nonsymmetric matrix $\mathbf{G}$. Thus, the condition number is not a precise metric to characterize the convergence. If these eigenvalues are clustered, then our bound can be small. On the other hand, if these eigenvalues are evenly distributed in the complex plane, then the bound can very close to 1.

More importantly, we would like to stress several technical contributions.

**1** : Our obtained Theorem 4.1 and 4.2 provide nonasymptotic guarantees, while most other work are asymptotic. For example, EG with positive momentum can achieve a asymptotic rate of $1 - O(1/\sqrt{\kappa})$ under strong assumptions (Azizian et al., 2020).

**2** : Our contribution is not just about fix the convergence issue of GDA by applying Anderson Mixing; another contribution is that we arrive at a convergent and tight bound on the original work and not just adopting existing analyses. We developed Theorem 4.1 and 4.2 from a new perspective because applying existing theoretical results fail to give us neither convergent nor tight bounds.

**3** : Theorem 4.1 and 4.2 only requires mild conditions and reflects how the table size $p$ controls the convergence rate. Theorem 4.1 is independent of the learning rate $\eta$. However, the convergence results of other methods like EG and OG depend on the learning rate, which may yield less than desirable results for ill-specified learning rates.

## 5 EXPERIMENTS

In this section, we conduct experiments to see whether `GDA-AM` improves GDA for minimax optimization from simple to practical problems. We first investigate performance of `GDA-AM` on bilinear games. In addition, we evaluate the efficacy of our approach on GANs.

## 5.1 BILINEAR PROBLEMS

In this section, we answer following questions: **Q1:** How is `GDA-AM` perform in terms of iteration number and running time? **Q2:** How is the scalability of `GDA-AM`? **Q3:** How is the performance of `GDA-AM` using different table size $p$? **Q4:** Does `GDA-AM` converge for large step size $\eta$?

We compare the performance with SimGDA, AltGDA, EG, and OG, and EG with Negative Momentum(Azizian et al. (2020)) on bilinear minimax games shown in equation 7 without any constraint.

, $\mathbf{b}, \mathbf{c}$, and initial points are generated using normally distributed random number. We set the maximum iteration number as $1 \times 10^6$, stopping criteria $1 \times 10^{-5}$ and depict convergence by use of the norm of distance to optima, which is defined as $\|\mathbf{w}^* - \mathbf{w}_t\|$. Similar to Azizian et al. (2020); Wei et al. (2021a), the step size is set as 1 after rescaling to have 2-norm 1. We present results of different settings in Figures 4, 5, and 6.

We first generate different problem size ($n = 100, 1000, 5000$) and present results of convergence in terms of iteration number in Figure 4. It can be observed that `GDA-AM` converges in much fewer iterations for different problem sizes. Note that EG, EG-NM, and OG converge in the end but requires many iterations, thus we plot only a portion for illustrative purposes. Figure 5 depicts the convergence for all methods in terms of time. It can be observed that the running time of `GDA-AM` is faster than EG. Although slower than OG, we can observe `GDA-AM` converges in much less time for all problems. Figure 4 and Figure 5 answer Q1 and Q2; although there is additional computation for `GDA-AM`, it does not hinder the benefits of adopting Anderson Mixing. Even for a large problem size, `GDA-AM` still converges in much less time than the baselines.

Next, we run `GDA-AM` using different table size $p$ and show the results in Figure 6a and Figure 6b. Figure 6a indicates an increasing of table size results in faster convergence in terms of iteration number, which also verifies our claim in Theorem 4.1. However, we also observe an increased running time when using a larger table size in Figure 6b. Further, we can see that $p = 50$ converges in a comparable time and iterations to $p = 100$. Similar results are found in repeated experiments as well. As a result, our answer to Q3 is that although a larger $p$ means less iterations, a medium $p$ is sufficient and a small $p$ still outperforms the baselines. The optimal choice of $p$ is related to the condition number and step size, which is another interesting topic in the Anderson Mixing community.

Next, we answer Q4 on convergence under different step sizes. Although `GDA-AM` usually converges with suitable step size, our theorem suggests it requires a larger table size when combined with a extremely aggressive step size. Figure 6c shows the convergence under such circumstance. We can observe that although a very large step size goes the wrong way in the beginning, Anderson Mixing can still make it back on track except when $\eta > 1$. It answers the question and confirms our claim that `GDA-AM` can achieve global convergence for bilinear problems for a large step size $\eta > 0$.

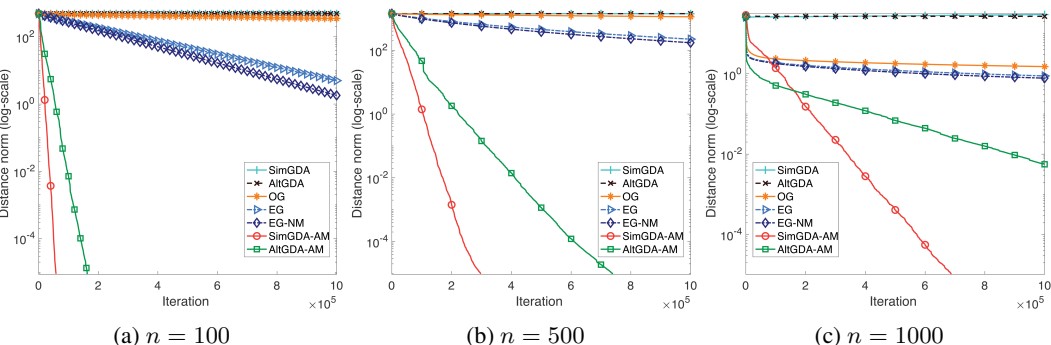

| (a) $n = 100$ | (b) $n = 500$ | (c) $n = 1000$ |

Figure 4: Comparison in terms of iteration: $\min_{\mathbf{x}} \max_{\mathbf{y}} f(\mathbf{x}, \mathbf{y}) = \mathbf{x}^T \mathbf{A} \mathbf{y} + \mathbf{b}^T \mathbf{x} + \mathbf{c}^T \mathbf{y}$. We use different problem size and fix $p = 10, \eta = 1$ for all experiments.

## 5.2 GAN EXPERIMENTS: IMAGE GENERATION

We apply our method to the CIFAR10 dataset (Krizhevsky, 2009) and use the ResNet architecture with WGAN-GP (Gulrajani et al., 2017) and SNGAN (Miyato et al., 2018) objective. We also compared

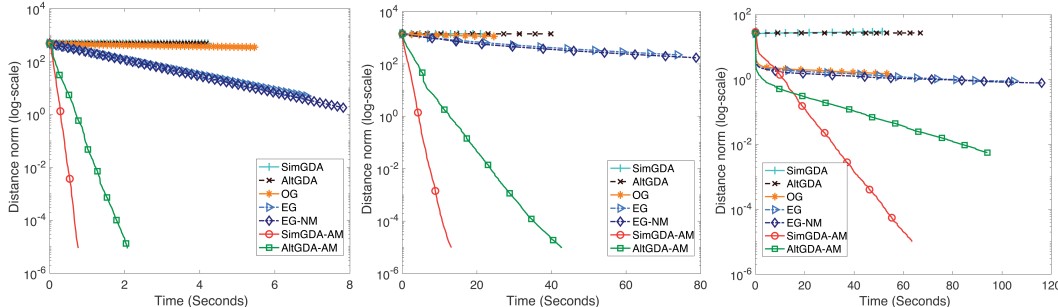

(a) Time comparison for Figure 4a    (b) Time comparison for Figure 4b    (c) Time comparison for Figure 4c

Figure 5: Comparison between methods in terms of time.

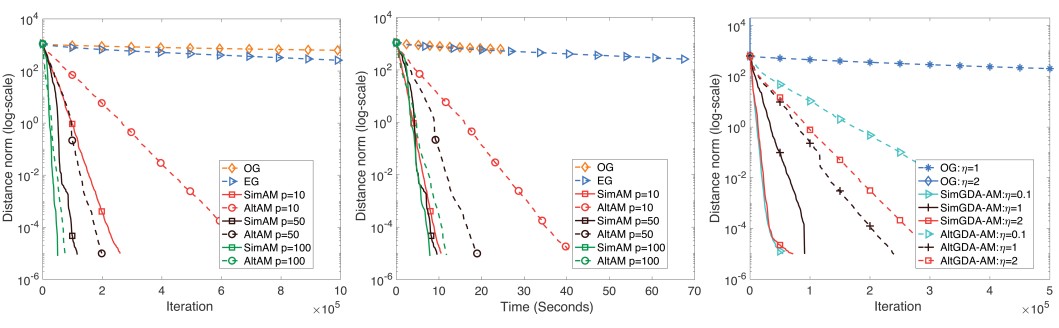

(a) Effects of $p$ in terms of iteration    (b) Time compasion for Figure 6a    (c) Effect of step size $\eta$, $p = 10$

Figure 6: Effects of table size $p$ and step size $\eta$, $n = 500$

the performance of GDA-AM using cropped CelebA (64×64) (Liu et al., 2015) on WGAN-GP. We compare with Adam and extra-gradient with Adam (EG) as it offers significant improvement over OG. Models are evaluated using the inception score (IS) (Salimans et al., 2016) and FID (Heusel et al., 2017) computed on 50,000 samples. For fair comparison, we fixed the same hyperparamters of Adam for all methods after an extensive search. Experiments were run with 5 random seeds. We show results in Table 1. Table 1 reports the best IS and FID (averaged over 5 runs) achieved on these datasets by each method. We see that GDA-AM yields improvements over the baselines in terms of generation quality.

Table 1: Best inception scores and FID for Cifar10 and FID for CelebA (IS is a less informative metric for celebA).

|  | WGAN-GP(ResNet) | | | SNGAN(ResNet) | |
|  | CIFAR10 | | CelebA | CIFAR10 | |
| Method | IS ↑ | FID ↓ | FID | IS | FID |
| Adam | 7.76 ±.11 | 22.45 ±.65 | 8.43 ±.05 | 8.21 ±.05 | 20.81 ±.16 |
| EG | 7.83 ±.08 | 20.73 ±.22 | 8.15 ±.06 | 8.15 ±.07 | 21.12 ±.19 |
| Ours (GDA-AM) | **8.05** ±.06 | **19.32** ±.16 | **7.82** ±.06 | **8.38** ±.04 | **18.84** ±.13 |

## 6 CONCLUSION

We prove the convergence property of GDA-AM and obtain a faster convergence rate than EG and OG on the bilinear problem. Empirically, we verify our claim for such a problem and show the efficacy of GDA-AM in a deep learning setting as well. We believe our work is different from previous approaches and takes an important step towards understanding and improving minimax optimization by exploiting the GDA dynamic and reforming it with numerical techniques.

ACKNOWLEDGMENTS

This work was funded in part by the NSF grant OAC 2003720, IIS 1838200 and NIH grant 5R01LM013323-03,5K01LM012924-03.

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

## A   RELATED WORK

There is a rich literature on different strategies to alleviate the issue of minimax optimization. A useful add-on technique, Momentum, has been shown to be effective for bilinear games and strongly-convex-strongly-concave settings (Zhang & Wang, 2021; Gidel et al., 2019b; Azizian et al., 2020). Several second-order methods (Adolphs et al.; Mescheder et al., 2017; Mazumdar et al., 2019; Parker-Holder et al., 2020) show that their stable fixed points are exactly either Nash equilibria or local minimax by incorporating second-order information. However, such methods are computationally expensive and thus unsuitable for large applications such as image generation. Focusing on variants of GDA, EG and OG are two widely studied algorithms on improving the GDA dynamics. EG proposed to apply extra-gradient to overcome the cycling behaviour of GDA. OG, originally proposed in Popov (1980) and rediscovered in Daskalakis et al. (2018); Mertikopoulos et al. (2019), is more efficient by storing and re-using the extrapolated gradient for the extrapolation step. Without projection, OG is equivalent to extrapolation from past. Mokhtari et al. (2020b) shows that both of these algorithms can be interpreted as approximations of the classical proximal point method and did a unified analysis for bilinear games. These approaches mentioned the GDA dynamics can be viewed as a fixed-point iteration, but none of them further provides a solution to improve it. In this work, we fill this gap by proposing the application of the extrapolation method directly on the entire GDA dynamics. Unlike OG, EG and their variants (Hsieh et al., 2019; Lei et al., 2021; Thekumparampil et al., 2019; Yang et al., 2019), which regard minimax problems as variational inequality problems (Bruck, 1977; Nemirovski, 2004), our work is from a new perspective and thus orthogonal to these previous approaches.

In addition, several recent works consider nonconvex-concave minimax problems. Zhang et al. (2020) introduced a "smoothing" scheme combined with GDA to stabilize the dynamic of GDA. Luo et al. (2020) proposed a method called Stochastic Recursive gradiEnt Descent Ascent (SREDA) for stochastic nonconvex-strongly-concave minimax problems, by estimating gradients recursively and reducing its variance. Lin et al. (2020) showed that the two-timescale GDA can find a stationary point of nonconvex-concave minimax problems effectively. Ostrovskii et al. (2021) proposed a variant of Nesterov's accelerated algorithm to find $\epsilon$-first-order Nash equilibrium that is a stronger criterion than the commonly used proximal gradient norm. Nouiehed et al. (2019) proposed a iterative method that finds $\epsilon$-first-order Nash equilibrium in $O(\epsilon^{-2})$ iterations under Polyak-Lojasiewicz (PL) condition. Focusing on nonconvex minimax problems, they studied an interesting and difficult problem. Since our work cast insight on the effectiveness of solving minimax optimization via Anderson Mixing, we expect the extension of this algorithm to general nonconvex problems can be further investigated in the future.

## B   ANDERSON MIXING IMPLEMENTATION DETAILS

In this section, we discuss the efficient implementation of Anderson Mixing. We start with generic Anderson Mixing prototype (Algorithm 4) and then present the idea of Quick QR-update Anderson Mixing implementation as described in Walker & Ni (2011b), which is commonly used in practice. For each iteration $t \geq 0$, AM prototype solves a least squares problem with a normalization constraint. The intuition is to minimize the norm of the weighted residuals of the previous $m$ iterates.

---

**Algorithm 4:** Anderson Mixing Prototype (truncated version)

---

**Input:** Initial point $w_0$, Anderson restart dimension $p$, fixed-point mapping $g : \mathbf{R}^n \to \mathbf{R}^n$.
**Output:** $w_{t+1}$
**for** *t = 0, 1, . . .* **do**
    Set $p_t = \min\{t, p\}$.
    Set $F_t = [f_{t-p_t}, \ldots, f_t]$, where $f_i = g(w_i) - w_i$ for each $i \in [t - p_t \mathinner{.\,.} t]$.
    Determine weights $\beta = (\beta_0, \ldots, \beta_{p_t})^T$ that solves $\min_\beta \|F_t\beta\|_2$, s. t. $\sum_{i=0}^{p_t} \beta_i = 1$.
    Set $w_{t+1} = \sum_{i=0}^{p_t} \beta_i g(w_{t-p_t+i})$.
**end**

---

The constrained linear least-squares problem in Algorithm AA can be solved in a number of ways. Our preference is to recast it in an unconstrained form suggested in Fang & Saad (2009); Walker & Ni (2011b) that is straightforward to solve and convenient for implementing efficient updating of QR.

Define $f_i = g(w_i) - w_i$, $\triangle f_i = f_{i+1} - f_i$ for each $i$ and set $F_t = [f_{t-p_t}, \ldots, f_t]$, $\mathcal{F}_t = [\triangle f_{t-p_t}, \ldots, \triangle f_t]$. Then solving the least-squares problem $(\min_\beta \|F_t\beta\|_2$, s. t. $\sum_{i=0}^{p_t} \beta_i = 1)$ is equivalent to

$$\min_{\gamma=(\gamma_0,\ldots,\gamma_{p_t-1})^T} \|f_t - \mathcal{F}_t\gamma\|_2 \tag{10}$$

where $\alpha$ and $\gamma$ are related by $\alpha_0 = \gamma_0, \alpha_i = \gamma_i - \gamma_{i-1}$ for $1 \le i \le p_t - 1$, and $\alpha_{p_t} = 1 - \gamma_{p_t-1}$.

Now the inner minimization subproblem can be efficiently solved as an unconstrained least squares problem by a simple variable elimination. This unconstrained least-squares problem leads to a modified form of Anderson Mixing

$$w_{t+1} = g(w_t) - \sum_{i=0}^{p_t-1} \gamma_i^{(t)} [g(w_{t-p_t+i+1}) - g(w_{t-p_t+i})] = g(w_t) - \mathcal{G}_t\gamma^{(t)}$$

where $\mathcal{G}_t = [\triangle g_{t-p_t}, \ldots, \triangle g_{t-1}]$ with $\triangle g_i = g(w_{i+1}) - g(w_i)$ for each $i$.

To obtain $\gamma^{(t)} = \left(\gamma_0^{(t)}, \ldots, \gamma_{p_t-1}^{(t)}\right)^T$ by solving equation 10 efficiently, we show how the successive least-squares problems can be solved efficiently by updating the factors in the QR decomposition $\mathcal{F}_t = Q_t R_t$ as the algorithm proceeds. We assume a think QR decomposition, for which the solution of the least-squares problem is obtained by solving the $p_t \times p_t$ linear system $R\gamma = Q' * f_t$. Each $\mathcal{F}_t$ is $n \times p_t$ and is obtained from $\mathcal{F}_{t-1}$ by adding a column on the right and, if the resulting number of columns is greater than $p$, also cleaning up (re-initialize) the table. That is, we never need to delete the left column because cleaning up the table stands for a restarted version of AM. As a result, we only need to handle two cases; 1 the table is empty(cleaned). 2 the table is not full. When the table is empty, we initialize $\mathcal{F}_1 = Q_1 R_1$ with $Q_1 = \triangle f_0 / \|\triangle f_0\|_2$ and $R = \|\triangle f_0\|_2$. If the table size is smaller than $p$, we add a column on the right of $\mathcal{F}_{t-1}$. Have $\mathcal{F}_{t-1} = QR$, we update $Q$ and $R$ so that $\mathcal{F}_t = [\mathcal{F}_{t-1}, \Delta f_{t-1}] = QR$. It is a single modified Gram–Schmidt sweep that is described as follows:

**Algorithm 5:** QR-updating procedures

---

**for** $i = 1, \ldots, p_{t-1}$ **do**
  Set $R(i, p_t) = Q(:, i)' * \triangle f_{t-1}$.
  Update $\triangle f_{t-1} \leftarrow \Delta f_{t-1} - R(i, p_t) * Q(:, i)$
**end**
Set $Q(:, p_t) = \triangle f_{t-1} / \|\triangle f_{t-1}\|_2$ and $R(p_t, p_t) = \|\Delta f_{t-1}\|_2$

---

Note that we do not explicitly conduct QR decomposition in each iteration, instead we update the factors ($O(p^2 n)$) and then solve a linear system using back substitution which has a complexity of $O(p^2)$. Based on this complexity analysis, we can find Anderson Mixing with QR-updating scheme has limited computational overhead than GDA (or OG). This explains why `GDA-AM` is faster than EG but slower than OG in terms of running time of each iteration.

## C  THEORETICAL RESULTS

### C.1  DIFFICULTY OF ANALYSIS ON GDA WITH ANDERSON MIXING

In the analysis, we study the inherent structures of the dynamics of the fixed point iteration and provide the convergence analysis for both simultaneous and alternating schemes. We want to emphasize that the direct application of existing convergence results of GMRES can not lead to convergent results. A recent paper Bollapragada et al. (2018) study the convergence acceleration schemes for multi-step optimization algorithms using Regularized Nonlinear Acceleration. We also want to point out that a naïve application of Crouzeix's bound to the minimax optimization problem can not be used to derive the convergent result.

**Theorem C.1** (Fischer & Freund (1991)). *Let $n \ge 5$ be an integer, $r > 1$, and $c \in \mathbb{R}$. Consider the following constrained polynomial minmax problem*

$$\min_{p \in \mathbb{P}_n : p(c)=1} \max_{z \in \mathscr{E}_r} |p(z)| \tag{11}$$

*where*

$$\mathscr{E}_r := \left\{ z \in \mathbb{C} \;\middle|\; |z-1| + |z+1| \le r + \frac{1}{r} \right\} \tag{12}$$

*and $c \in \mathbb{C} \setminus \mathscr{E}_r$. Then this problem can be solved uniquely by*

$$t_n(z; c) := \frac{T_n(z)}{T_n(c)}, \tag{13}$$

*where*

$$T_n(z) = \frac{1}{2} \left( v^n + \frac{1}{v^n} \right), \quad z = \frac{1}{2} \left( v + \frac{1}{v} \right) \tag{14}$$

*if*
*(a) $|c| \geq \frac{1}{2} \left( r^{\sqrt{2}} + r^{-\sqrt{2}} \right)$ or*
*(b) $|c| \geq (1/2a_r) \left( 2a_r^2 - 1 + \sqrt{2a_r^4 - a_r^2 + 1} \right)$, where $a_r := \frac{1}{2} \left( r + \frac{1}{r} \right)$.*

This is because the point $0$ where all the residual polynomials take the fixed value of $1$ is included in the numerical range of the iteration matrix, which violates the assumption of Theorem C.1. As a result, it can not be used to prove that the residual norm is decreasing based on this approach. Instead, we show that although the coefficient matrix is non-normal, it is diagonalizable. We then give the convergence results based on the eigenvalues instead of the numerical range. More specifically, Anderson mixing is equivalent to GMRES being applied to solve the following linear system:

$$(\mathbf{I} - \mathbf{G}^{(Alt)})\mathbf{w} = \mathbf{b}^{(Alt)}, \quad \text{with } \mathbf{w}_0 = \mathbf{w}_0^{(Alt)}. \tag{15}$$

Writing this linear system in the block form:

$$\begin{bmatrix} \mathbf{0} & \eta\mathbf{A} \\ -\eta\mathbf{A}^T & \eta^2\mathbf{A}^T \end{bmatrix} \mathbf{w} = \mathbf{b}^{(Alt)}. \tag{16}$$

The residual norm bound for GMRES reads:

$$\|\mathbf{r}_t\|_2 = \min_{p \in \mathbb{P}_t^1} \|p(\mathbf{I} - \mathbf{G}^{(Alt)})\mathbf{r}_0\|_2. \tag{17}$$

Notice that the matrix $(\mathbf{I} - \mathbf{G}^{(Alt)})$ is non-normal. If we apply Crouzeix's bound in Crouzeix & Palencia (2017) to our problem as Bollapragada et al. (2018) did, then we have the following bound

$$\frac{\|\mathbf{r}_t\|_2}{\|\mathbf{r}_0\|_2} \leq \min_{p \in \mathbb{P}_t^1} \|p(\mathbf{I} - \mathbf{G}^{(Alt)})\| \leq (1 + \sqrt{2}) \min_{p \in \mathbb{P}_t^1} \sup_{z \in W(\mathbf{I} - \mathbf{G}^{(Alt)})} \|p(z)\| \tag{18}$$

where $W(\mathbf{I} - \mathbf{G}^{(Alt)}) = \{\mathbf{z}^*(\mathbf{I} - \mathbf{G}^{(Alt)})\mathbf{z}, \forall \mathbf{z} \in \mathbb{C}^{2n} \setminus \{\mathbf{0}\}, \|z\| = 1\}$ is the numerical range for $\mathbf{I} - \mathbf{G}^{(Alt)}$. In order to simplify the upper bound in the previous theorem, we study the numerical range of $\mathbf{I} - \mathbf{G}^{(Alt)}$ similar to Bollapragada et al. (2018). Writing $\mathbf{z} = \begin{bmatrix} \mathbf{z}_1 \\ \mathbf{z}_2 \end{bmatrix}$ and computing the numerical range of $\mathbf{I} - \mathbf{G}^{(Alt)}$ explicitly yields:

$$[\mathbf{z}_1^*, \mathbf{z}_2^*] \begin{bmatrix} \mathbf{0} & \eta\mathbf{A} \\ -\eta\mathbf{A}^T & \eta^2\mathbf{A}^T \end{bmatrix} \begin{bmatrix} \mathbf{z}_1 \\ \mathbf{z}_2 \end{bmatrix} = \eta^2\mathbf{z}_2^*\mathbf{A}^T\mathbf{z}_2 + \eta\mathbf{z}_1^*\mathbf{z}_2 - \eta\mathbf{z}_2^{*T}\mathbf{z}_1. \tag{19}$$

For a general matrix $A$, there is no special structure about the numerical range of $\mathbf{I} - \mathbf{G}^{(Alt)}$. However, when is symmetric, we can decompose as $= \sum_{i=1}^n \lambda_i \mathbf{v}_i \mathbf{v}_i^T$ where $\{\lambda_i\}_{i=1}^n$ are eigenvalues of in decreasing order and $\{\mathbf{v}_i\}_{i=1}^n$ are associated eigenvectors, and write $\mathbf{A}^T = \sum_{i=1}^n \lambda_i^2 \mathbf{v}_i \mathbf{v}_i^T$. Then we can compute the numerical range of $\mathbf{G}^{(Alt)}$ as follows:

$$\sum_i^n [\mathbf{z}_1^*, \mathbf{z}_2^*] \begin{bmatrix} \mathbf{0} & \eta\lambda_i \mathbf{v}_i \mathbf{v}_i^T \\ -\eta\lambda_i \mathbf{v}_i \mathbf{v}_i^T & \eta^2\lambda_i^2 \mathbf{v}_i \mathbf{v}_i^T \end{bmatrix} \begin{bmatrix} \mathbf{z}_1 \\ \mathbf{z}_2 \end{bmatrix} = \sum_i^n [\mathbf{z}_1^*\mathbf{v}_i, \mathbf{z}_2^*\mathbf{v}_i] \begin{bmatrix} 0 & \eta\lambda_i \\ -\eta\lambda_i & \eta^2\lambda_i^2 \end{bmatrix} \cdot \begin{bmatrix} \mathbf{v}_i^T \mathbf{z}_1 \\ \mathbf{v}_i^T \mathbf{z}_2 \end{bmatrix} \tag{20}$$

Following the techniques proposed in Bollapragada et al. (2018) to analyze the numerical range of general $2 \times 2$ matrices, we can show that the numerical range of $\mathbf{I} - \mathbf{G}^{(Alt)}$ is equal to the convex hull of the union of the numerical range of

$$\mathbf{G}_i = \begin{bmatrix} 0 & \eta\lambda_i \\ -\eta\lambda_i & \eta^2\lambda_i^2 \end{bmatrix}, i = 1, \dots, n. \tag{21}$$

And the boundary of numerical range of $\mathbf{G}_i$ is an ellipse whose axes are the line segments joining the points x to y and w to z, respectively, with

$$x = 0, \quad y = \eta^2 \lambda_i^2, \quad , w = \frac{\eta^2 \lambda_i^2}{2} - \sqrt{-1}\eta|\lambda_i|, \quad z = \frac{\eta^2 \lambda_i^2}{2} + \sqrt{-1}\eta|\lambda_i|. \tag{22}$$

Thus, the numerical range of $\mathbf{I} - \mathbf{G}^{(Alt)}$ can be spanned by convex hull of the union of the numerical range of a set of 2-by-2 matrices and the numerical range of each such a 2-by-2 matrix is an ellipse. We can compute the center o and focal distance d of the ellipse generated by numerical range of $\mathbf{I} - \mathbf{G}^{(Alt)}$ explicitly. Then a linear transformation enables us to use Theorem C.1 to show that the near-best polynomial for the minimax problem on the numerical range of $\mathbf{I} - \mathbf{G}^{(Alt)}$ is given by $t_n(z; c) := \frac{T_n(\frac{z-o}{d})}{T_n(\frac{c-o}{d})}$ if 0 is excluded from the numerical range of $\mathbf{I} - \mathbf{G}^{(Alt)}$. However, according to equation 22 the numerical range includes the point 0 where the residual polynomial takes value 1, thus the analysis based on numerical range can not help derive the convergent result as the upper bound is not guaranteed to be less than 1.

## C.2 Proofs of theorem

We first provide proof of Theorem 4.1.

**Theorem C.2** (Global convergence for simultaneous GDA-AM on bilinear problem). *Denote the distance between the stationary point* $\mathbf{w}^*$ *and current iterate* $\mathbf{w}_{(k+1)p}$ *of Algorithm 2 with Anderson restart dimension p as* $N_{(k+1)p} = dist(\mathbf{w}^*, \mathbf{w}_{(k+1)p})$. *Then we have the following bound for* $N_t$ *Algorithm 2 is unconditionally convergent*

$$N_{(k+1)p} \leq \frac{1}{T_p(1 + \frac{2}{\kappa(T)-1})} N_{kp} \tag{23}$$

*where* $T_p$ *is the Chebyshev polynomial of first kind of degree p and* $\frac{1}{T_p(1+\frac{2}{\kappa(T)-1})} < 1$ *since* $1 + \frac{2}{\kappa(T)-1} > 1$.

***Proof of Theorem 4.1.*** Note that $\mathbf{I} - \mathbf{G}^{(Sim)}$ is a normal matrix which will be denoted as $\mathbf{G}$ for notational simplicity. Thus it admits the following eigendecomposition:

$$\mathbf{G} = \mathbf{U}\mathbf{\Lambda}\mathbf{U}^T, \quad \mathbf{U}\mathbf{U}^T = \mathbf{I}, \quad \Lambda = \text{diag}(\lambda_1, \ldots, \lambda_{2n}). \tag{24}$$

Based on the equivalence between GMRES and Anderson Mixing, we know that the convergence rate of simultaneous GDA-AM can be estimated by the spectrum of $\mathbf{G}$. Especially, it holds that

$$\mathbf{r}_{(k+1)p} = \mathbf{U}f_p(\mathbf{\Lambda})\mathbf{U}^T\mathbf{r}_{kp}. \quad f_p \in \mathcal{P}_p \tag{25}$$

where $\mathcal{P}_p$ is the family of residual polynomials with degree p such that $f_p(0) = 1, \forall f_p \in \mathcal{P}_p$. According to Lemma 2.1, we have the following estimation

$$\|\mathbf{r}_{(k+1)p}\|_2 = \min_{f_p \in \mathcal{P}_p} \|f_p(\mathbf{G})\mathbf{r}_{kp}\|_2 \leq \min_{f_p \in \mathcal{P}_p} \max_i |f_p(\lambda_i)| \|\mathbf{r}_{kp}\|_2. \tag{26}$$

Due to the block structure of $\mathbf{G}$, the eigenvalues of $\mathbf{G}$ can be computed explicitly as

$$\pm \eta\sigma_i\sqrt{-1}, \ i = 1, \ldots, n, \tag{27}$$

where $\sigma_i$ is the $i$th largest singular value of matrix . This shows that the eigenvalues of $\mathbf{G}$ are $n$ pairs of purely imaginary numbers excluding 0 since has full rank.

Since the eigenvalues of $\mathbf{G}$ are distributed in two intervals excluding the origin

$$I = [-\eta\sigma_{max}()\sqrt{-1}, -\eta\sigma_{min}()\sqrt{-1}] \cup [\eta\sigma_{min}()\sqrt{-1}, \eta\sigma_{max}()\sqrt{-1}],$$

it can be shown that the following p-th degree polynomial with value 1 at the origin that has the minimal maximum deviation from 0 on I is given by:

$$f_p(z) = \frac{T_l(q(\sqrt{-1}z))}{T_l(q(0))}, \quad q(\sqrt{-1}z) = 1 - \frac{2(\sqrt{-1}z - \eta\sigma_{min})(\sqrt{-1}z + \eta\sigma_{min})}{(\eta\sigma_{max}())^2 - (\eta\sigma_{min}())^2} \tag{28}$$

where $l = \lceil \frac{p}{2} \rceil$ and $T_l$ is the Chebyshev polynomial of first kind of degree $l$. The function $q(\sqrt{-1}z)$ maps I to $[-1, 1]$. Thus the numerator of the polynomial $f_p$ is bounded by 1 on I. The size of denominator can be determined by the method discussed in Chapter 3 of Greenbaum (1997). Assume $q(0) = \frac{1}{2}(y + y^{-1})$, then $T_l(q(0)) = \frac{1}{2}(y^l + y^{-l})$. Then y can be determined by solving

$$q(0) = \frac{(\eta\sigma_{max}())^2 + (\eta\sigma_{min}())^2}{(\eta\sigma_{max}())^2 - (\eta\sigma_{min}())^2}. \tag{29}$$

The solutions to this equation are

$$y_1 = \frac{\eta\sigma_{max}() + \eta\sigma_{min}()}{\eta\sigma_{max}() - \eta\sigma_{min}()} \quad \text{or} \quad y_2 = \frac{\eta\sigma_{max}() - \eta\sigma_{min}()}{\eta\sigma_{max}() + \eta\sigma_{min}()}. \tag{30}$$

Then plugging the value of $q(0)$ into the polynomial $f_p$ yields

$$\frac{\|\mathbf{r}_{(k+1)p}\|}{\|\mathbf{r}_{kp}\|} \le 2\Big(\frac{\sqrt{\eta^2\sigma_{max}^2()} - \sqrt{\eta^2\sigma_{min}^2()}}{\sqrt{\eta^2\sigma_{max}^2()} + \sqrt{\eta^2\sigma_{min}^2()}}\Big)^l$$

$$= 2\Big(\frac{\sigma_{max}() - \sigma_{min}()}{\sigma_{max}() + \sigma_{min}()}\Big)^l = 2\Big(\frac{\kappa() - 1}{\kappa() + 1}\Big)^l \tag{31}$$

Note that $N_t$ and $\mathbf{r}_t$ is related through $\mathbf{G}(\mathbf{w}_t - \mathbf{w}^*) = \mathbf{r}_t$. Therefore,

$$N_{(k+1)p} = \|\mathbf{w}_{(k+1)p} - \mathbf{w}^*\|_2 = \|\mathbf{G}^{-1}\mathbf{r}_{(k+1)p}\|_2 = \min_{f_p \in \mathcal{P}_p} \|\mathbf{G}^{-1}f_p(\mathbf{G})\mathbf{G}(\mathbf{w}_{kp} - \mathbf{w}^*)\|_2$$

$$\le \min_{f_p \in \mathcal{P}_p} \max_i |f_p(\lambda_i)|\|\mathbf{w}_{kp} - \mathbf{w}^*\|_2 \le 2\Big(1 - \frac{2}{\kappa() + 1}\Big)^{\frac{p}{2}} N_{kp}. \tag{32}$$

Actually a tighter bound can be proved after noting that the problem is essentially equivalent to polynomial minmax problem on the interval:

$$\tilde{I} = [\eta^2\sigma_{min}^2(), \ \eta^2\sigma_{max}^2()],$$

Then it is well known that,

$$N_{(k+1)p} \le \min_{f_p \in \mathcal{P}_p} \max_{\lambda_i \in [\eta^2\sigma_{min}^2(), \ \eta^2\sigma_{max}^2()]} |f_p(\lambda_i)|\|\mathbf{w}_{kp} - \mathbf{w}^*\|_2 \le \frac{1}{T_p(1 + 2\frac{\sigma_{min}^2}{\sigma_{max}^2 - \sigma_{min}^2})} N_{kp}$$

$$\le \frac{1}{T_p(1 + \frac{2}{\kappa(T) - 1})} N_{kp} \tag{33}$$

where $T_p$ Chebyshev polynomial of degree p of the first kind and $\frac{1}{T_p(1 + \frac{2}{\kappa(T) - 1})} < 1$. Explicitly,

$$T_p(1 + \frac{2}{\kappa(T) - 1}) = \frac{1}{2}\Big[\Big(1 + \frac{2}{\kappa(T) - 1} + \sqrt{(1 + \frac{2}{\kappa(T) - 1})^2 - 1}\Big)^p$$

$$+ \Big(1 + \frac{2}{\kappa(T) - 1} + \sqrt{(1 + \frac{2}{\kappa(T) - 1})^2 - 1}\Big)^{-p}\Big]$$

□

Next, we give the proof of Theorem 4.2.

**Theorem C.3** (Global convergence for alternating `GDA-AM` on bilinear problem). *Denote the distance between the stationary point $\mathbf{w}^*$ and current iterate $\mathbf{w}_{(k+1)p}$ of Algorithm 3 with Anderson restart dimension p as $N_{(k+1)p} = dist(\mathbf{w}^*, \mathbf{w}_{(k+1)p})$. Assume is normalized such that its largest singular value is equal to 1. Then when the learning rate $\eta$ is less than 2, we have the following bound for $N_t$*

$$N_{(k+1)p}^2 \le \sqrt{1 + \frac{2\eta}{2 - \eta}}(\frac{r}{c})^p N_{kp}^2$$

*where c and r are the center and radius of a disk $D(c, r)$ which includes all the eigenvalues of $\mathbf{G}$ in equation 4.3. Especially, $\frac{r}{c} < 1$.*

*Proof.* Since the residual $\mathbf{r}_p$ of AA at p-th iteration has the form of

$$\mathbf{r}_p = (\mathbf{I} - \sum_{i=1}^{p} \mathbf{G}^i)\mathbf{r}_0,$$

and AA minimizes the residual, we have

$$\|\mathbf{r}_{(k+1)p}\|_2^2 \leq \min_{\beta} \|\mathbf{r}_{kp} - \beta \mathbf{G}^i \mathbf{r}_{kp}\|_2^2 \leq \min_{f_p \in \mathcal{P}_p} \|f_p(\mathbf{G})\mathbf{r}_{kp}\|_2^2,$$

where $\mathcal{P}_p$ is the family of polynomials with degree p such that $f_p(0) = 1, \forall f_p \in \mathcal{P}_p$ . It's easy to see that $\mathbf{G}$ is unitarily similar to a block diagonal matrix $\Lambda$ with $2 \times 2$ blocks as follows:

$$\begin{bmatrix} 0 & \eta\sigma_i \\ -\eta\sigma_i & (\eta\sigma_i)^2 \end{bmatrix} \quad \forall\, i \in [n].$$

Thus the eigenvalues of $G$ can be easily identified as

$$\lambda_{\pm i} = \frac{(\eta\sigma_i(\eta\sigma_i \pm \sqrt{(\eta\sigma_i)^2 - 4}))}{2}, \quad i \in [n].$$

where $\sigma_1 \geq \sigma_2 \geq \cdots \geq \sigma_n$ are the singular values of . Furthermore, the eigenvector and eigenvalue associated with each $2 \times 2$ diagonal block are

$$\begin{bmatrix} 0 & \eta\sigma_i \\ -\eta\sigma_i & (\eta\sigma_i)^2 \end{bmatrix} \begin{bmatrix} 1 \\ \frac{\lambda_{\pm i}}{\eta\sigma_i} \end{bmatrix} = \lambda_{\pm i} \begin{bmatrix} 1 \\ \frac{\lambda_{\pm i}}{\eta\sigma_i} \end{bmatrix}$$

Thus $\mathbf{G}$ is diagonalizable and denote the matrix with the columns of eigenvectors of $\mathbf{G}$ by X. The real part of the eigenvalues of $\mathbf{G}$ are at least

$$\mathcal{R}(\lambda_{\pm i}) \geq \frac{(\eta\sigma_i)^2}{2}, \quad i \in [n]. \tag{34}$$

And since $|\eta\sigma_i| \geq |\sqrt{(\eta\sigma_i)^2 - 4}|$, all the eigenvalues will be included in a disk $D(c, r)$ which is included in the right half plane. Moreover, both c and r being greater than zero indicates that $\frac{r}{c} < 1$. Start from the following inequality:

$$N_{(k+1)p} = \left\|\mathbf{w}_{(k+1)p} - \mathbf{w}^*\right\|_2 = \left\|\mathbf{G}^{-1}\mathbf{r}_{(k+1)p}\right\|_2 \leq \min_{f_p \in \mathcal{P}_p} \left\|\mathbf{G}^{-1}f_p(\mathbf{G})\mathbf{r}_{kp}\right\|_2$$

$$= \min_{f_p \in \mathcal{P}_p} \left\|\mathbf{G}^{-1}f_p(\mathbf{G})\mathbf{G}\left(\mathbf{w}_{kp} - \mathbf{w}^*\right)\right\|_2 = \min_{f_p \in \mathcal{P}_p} \left\|\mathbf{G}_p^{-1}(\mathbf{G})\left(\mathbf{w}_{kp} - \mathbf{w}^*\right)\right\|_2 \tag{35}$$

$$= \min_{f_p \in \mathcal{P}_p} \left\|f_p(\mathbf{G})\left(\mathbf{w}_{kp} - \mathbf{w}^*\right)\right\|_2$$

We will use the eigendecomposition of $G$ and the special polynomial $(\frac{c-t}{c})^p$ to derive the inequality in Theorem 3. Now we know $\frac{r}{c} < 1$. If we choose $g_p(t) = (\frac{c-t}{c})^p$, we can obtain

$$\min_{f_p \in \mathcal{P}_p} \|f_p(\mathbf{G})(\mathbf{w}_{kp} - \mathbf{w}^*)\|_2 \leq \|g_p(\mathbf{G})(\mathbf{w}_{kp} - \mathbf{w}^*)\|_2$$

which implies

$$\min_{f_p \in \mathcal{P}_p} \|f_p(\mathbf{G})(\mathbf{w}_{kp} - \mathbf{w}^*)\|_2 \leq \|g_p(\mathbf{X}\Lambda\mathbf{X}^{-1})\|\|(\mathbf{w}_{kp} - \mathbf{w}^*)\|_2$$

Since $G$ is diagonalizable (which has been shown above), we assume the eigendecomposition of $\mathbf{G}$ is $\mathbf{G} = \mathbf{X}\Lambda\mathbf{X}^{-1}$. Then

$$\min_{f_p \in \mathcal{P}_p} \|g_p(\mathbf{G})(\mathbf{w}_{kp} - \mathbf{w}^*)\|_2 \leq \|\mathbf{X}\|\|\mathbf{X}^{-1}\| \max_{\{\lambda_i\}_{i=1}^{2n}} \|g_p(\Lambda)\|\|(\mathbf{w}_{kp} - \mathbf{w}^*)\|_2 \leq \kappa_{\mathbf{G}}(\frac{r}{c})^p\|\mathbf{w}_{kp} - \mathbf{w}^*\|_2$$

where $\kappa_{\mathbf{G}}$ is the condition number of $X$. The last inequality comes from Lemma 6.26 and Proposition 6.32 in Saad (2003).. Since $\mathbf{G}$ and $\Lambda$ are unitarily similar, $\kappa_{\mathbf{G}}$ is equal to the condition number of the eigenvector matrix of $\Lambda$. The eigenvector matrix of $\Lambda$ is a block diagonal matrix with the $i$th block as $\begin{bmatrix} 1 & 1 \\ \frac{\lambda_{+i}}{\eta\sigma_i} & \frac{\lambda_{-i}}{\eta\sigma_i} \end{bmatrix}$. Thus the singluar values of the eigenvector matrix of $\Lambda$ is equal to the union of the singular values of these 2-by-2 blocks. Under the assumption that the largest singular value of  are equal to 1 and the learning rate is less than 2, it is easy to find the singular values of the eigenvector matrix of $\Lambda$ are $\sqrt{2 \pm \eta\sigma_i}$. Thus, $\kappa_{\mathbf{G}} = \frac{\sqrt{2+\eta\sigma_{\max}}}{\sqrt{2-\eta\sigma_{\max}}} = \frac{\sqrt{2+\eta}}{\sqrt{2-\eta}} = \sqrt{1 + \frac{2\eta}{2-\eta}}$. $\qquad\square$

C.3 DISCUSSION OF OBTAINED RATES

We would like to first explain on why taking Chebyshev polynomial of degree p at the point $1 + \frac{2}{\kappa - 1}$. We evaluate the Chebyshev polynomial at this specific point because the reciprocal of this value gives the minimal value of infinite norm of the all polynomials of degree p defined on the interval $\tilde{I} = [\eta^2 \sigma_{min}^2(), \ \eta^2 \sigma_{max}^2()]$ based on Theorem 6.25 (page 209) (Saad, 2003). In other words, taking the function value at this point leads to the tight bound.

When comparing between existing bounds, we would like to point our our derived bounds are hard to compare directly. Alternatively, we can derive another bound for comparison with existing bounds for simultaneous `GDA-AM`. If we use the inequality that $T_p(t) \geq \frac{1}{2}((t + \sqrt{t^2 - 1})^p)$, we can obtain the bound $\rho(A) = 4(\frac{\sqrt{\kappa(A^T A)} - 1}{\sqrt{\kappa(A^T A)} + 1})^2 = 4(1 - O(\frac{1}{\sqrt{\kappa(A^T A)}}))$, which is in a form that is comparable with EG and can compete with EG + positive momentum. The numerical experiments in figure 2b numerically verify that our bound is smaller than EG. We wanted to numerically compare our rate with EG with positive momentum. However the bound of EG with positive momentum is asymptotic. Moreover, it does not specify the constants so we can not numerically compare them. We do provide empirical comparison between GDA-AM and EG with positive momentum for bilinear problems in Appendix D.1. It shows GDA-AM outperforms EG with positive momentum. Regarding alternating `GDA-AM` , we would like to note that the bound in Theorem 4.2 depends on the eigenvalue distribution of the matrix $\mathbf{G}$. Condition number is not directly related to the distribution of eigenvalues of a nonsymmetric matrix $\mathbf{G}$. Thus, the condition number is not a precise metric to characterize the convergence. If these eigenvalues are clustered, then our bound can be small. On the other hand, if these eigenvalues are evenly distributed in the complex plane, then the bound can very close to 1.

More importantly, we would like to stress several technical contributions.

1. Our obtained Theorem 4.1 and 4.2 provide nonasymptotic guarantees, while most other work are asymptotic. For example, EG with positive momentum can achieve a asymptotic rate of $1 - O(1/\sqrt{\kappa})$ under strong assumptions (Azizian et al., 2020).

2. Our contribution is not just about fix the convergence issue of GDA by applying Anderson Mixing; another contribution is that we arrive at a convergent and tight bound on the original work and not just adopting existing analyses. We developed Theorem 4.1 and 4.2 from a new perspective because applying existing theoretical results fail to give us neither convergent nor tight bounds.

3. Theorem 4.1 and 4.2 only requires mild conditions and reflects how the table size $p$ controls the convergence rate. Theorem 4.1 is independent of the learning rate $\eta$. However, the convergence results of other methods like EG and OG depend on the learning rate, which may yield less than desirable results for ill-specified learning rates.

C.4 CONVEX-CONCAVE AND GENERAL CASE

Given the widespread usage of minimax problems in applications of machine learning, it is natural to ask about its properties when being applied to general nonconvex-nonconcave settings. If $f$ is a nonconvex-nonconcave function, the problem of finding global Nash equilibrium is NP-hard in general. Recently, Jin et al. (2020) show that local or global Nash equilibrium may not exist in nonconvex-nonconcave settings and propose a new notation *local minimax* as defined below:

**Definition 4.** *A point* $(\mathbf{x}^\star, \mathbf{y}^\star)$ *is said to be a **local minimax** point of* $f$, *if there exists* $\delta_0 > 0$ *and a function* $h$ *satisfying* $h(\delta) \to 0$ *as* $\delta \to 0$, *such that for any* $\delta \in (0, \delta_0]$, *and any* $(\mathbf{x}, \mathbf{y})$ *satisfying* $\|\mathbf{x} - \mathbf{x}^\star\| \leq \delta$ *and* $\|\mathbf{y} - \mathbf{y}^\star\| \leq \delta$, *we have*

$$f(\mathbf{x}^\star, \mathbf{y}) \leq f(\mathbf{x}^\star, \mathbf{y}^\star) \leq \max_{\mathbf{y}': \|\mathbf{y}' - \mathbf{y}^\star\| \leq h(\delta)} f(\mathbf{x}, \mathbf{y}').$$

Jin et al. (2020) also establishes the following first- and second-order conditions to characterize local minimax:

**Proposition 1** (First-order Condition). *Any local minimax point* $(\mathbf{x}^*, \mathbf{y}^*)$ *satisfies* $\nabla f(\mathbf{x}^*, \mathbf{y}^*) = \mathbf{0}$.

**Proposition 2** (Second-order Necessary Condition). *Any local minimax point* $(\mathbf{x}^*, \mathbf{y}^*)$ *satisfies* $\nabla_{\mathbf{yy}} f(\mathbf{x}^*, \mathbf{y}^*) \preccurlyeq \mathbf{0}$ *and* $\nabla_{\mathbf{xx}} f(\mathbf{x}^*, \mathbf{y}^*) - \nabla_{\mathbf{xy}} f(\mathbf{x}^*, \mathbf{y}^*)(\nabla_{\mathbf{yy}} f(\mathbf{x}^*, \mathbf{y}^*))^{-1} \nabla_{\mathbf{yx}} f(\mathbf{x}^*, \mathbf{y}^*) \succcurlyeq \mathbf{0}$.

**Proposition 3** (Second-order Sufficient Condition). *Any stationary point $(\mathbf{x}^*, \mathbf{y}^*)$ satisfies $\nabla_{\mathbf{yy}} f(\mathbf{x}^*, \mathbf{y}^*) \prec \mathbf{0}$ and $\nabla_{\mathbf{xx}} f(\mathbf{x}^*, \mathbf{y}^*) - \nabla_{\mathbf{xy}} f(\mathbf{x}^*, \mathbf{y}^*)(\nabla_{\mathbf{yy}} f(\mathbf{x}^*, \mathbf{y}^*))^{-1} \nabla_{\mathbf{yx}} f(\mathbf{x}^*, \mathbf{y}^*) \succ \mathbf{0}$ is a local minimax point.*

Given the second-order conditions of local minimax, it turns out that above question is extremely challenging—`GDA-AM` is a first-order method. But we can prove the following result for `GDA-AM`:

**Theorem C.4** (Local minimax as subset of limiting points of `GDA-AM`). *Consider a general objective function $f(\mathbf{x}, \mathbf{y})$. The set of limiting points of `GDA-AM` for minimax problem*

$$\min_{\mathbf{x} \in \mathbb{R}^n} \max_{\mathbf{y} \in \mathbb{R}^n} f(\mathbf{x}, \mathbf{y})$$

*includes the local minimax points of this function.*

The definition of local minimax is stronger than that of first order $\epsilon$ point. The convergence analysis for complexity of finding $\epsilon$ stationary point is included in the next section. The proof of Theorem C.4 needs the result from the following theorem.

**Theorem C.5** (Calvetti et al. (2002)). *Let $\delta$ satisfy $0 < \delta \le \delta_0$ for some constant $\delta_0 > 0$ (refer to Calvetti et al. (2002) for details), and let $b^\delta \in \mathcal{X}$ satisfy $\left\| b - b^\delta \right\| \le \delta$. Let $k \le \ell$ and let $x_k^\delta$ denote the $k$th iterate determined by the GMRES method applied to equation $Ax = b^\delta$, with initial guess $x_0^\delta = 0$. Similarly, let $x_k$ denote the $k$th iterate determined by the GMRES method applied to equation $Ax = b$ with initial guess $x_0 = 0$. Then, there are constants $\sigma_k$ independent of $\delta$, such that*

$$\left\| x_k - x_k^\delta \right\| \le \sigma_k \delta, \quad 1 \le k \le \ell$$

Then, we give the proof of Theorem C.4.

***Proof of Theorem C.4.*** For notational simplicity, we will denote $\nabla_{\mathbf{xx}} f(\mathbf{x}^*, \mathbf{y}^*)$, $\nabla_{\mathbf{xy}} f(\mathbf{x}^*, \mathbf{y}^*)$ and $\nabla_{\mathbf{yy}} f(\mathbf{x}^*, \mathbf{y}^*)$ by $\mathbf{H}_{\mathbf{x}^*\mathbf{x}^*}$, $\mathbf{H}_{\mathbf{x}^*\mathbf{y}^*}$ and $\mathbf{H}_{\mathbf{y}^*\mathbf{y}^*}$, respectively. Simultaneous GDA can be written as

$$\mathbf{w}_{t+1} = \begin{bmatrix} \mathbf{x}_{t+1} \\ \mathbf{y}_{t+1} \end{bmatrix} = \begin{bmatrix} \mathbf{x}_t - \eta \nabla_{\mathbf{x}} f(\mathbf{x}_t, \mathbf{y}_t) \\ \mathbf{y}_t + \eta \nabla_{\mathbf{y}} f(\mathbf{x}_t, \mathbf{y}_t) \end{bmatrix}.$$

Since the function is differentiable, Taylor expansion holds for $\nabla_{\mathbf{x}} f(\mathbf{x}_t, \mathbf{y}_t)$ and $\nabla_{\mathbf{y}} f(\mathbf{x}_t, \mathbf{y}_t)$ at a local minimx point $\mathbf{w}^* = (\mathbf{x}^*, \mathbf{y}^*)$,

$$\nabla_{\mathbf{x}} f(\mathbf{x}_t, \mathbf{y}_t) = \nabla_{\mathbf{x}} f(\mathbf{x}^*, \mathbf{y}^*) + \mathbf{H}_{\mathbf{x}^*\mathbf{x}^*}(\mathbf{x}_t - \mathbf{x}^*) + \mathbf{H}_{\mathbf{x}^*\mathbf{y}^*}(\mathbf{y}_t - \mathbf{y}^*) + o(\|\mathbf{w}_t - \mathbf{w}^*\|_2)$$
$$\nabla_{\mathbf{y}} f(\mathbf{x}_t, \mathbf{y}_t) = \nabla_{\mathbf{y}} f(\mathbf{x}^*, \mathbf{y}^*) + \mathbf{H}_{\mathbf{y}^*\mathbf{y}^*}(\mathbf{y}_t - \mathbf{y}^*) + \mathbf{H}_{\mathbf{y}^*\mathbf{x}^*}(\mathbf{x}_t - \mathbf{x}^*) + o(\|\mathbf{w}_t - \mathbf{w}^*\|_2).$$

Use the fact that $\nabla f(\mathbf{x}^*, \mathbf{y}^*) = \mathbf{0}$ to simplify the above equations and obtain

$$\nabla_{\mathbf{x}} f(\mathbf{x}_t, \mathbf{y}_t) = \mathbf{H}_{\mathbf{x}^*\mathbf{x}^*}(\mathbf{x}_t - \mathbf{x}^*) + \mathbf{H}_{\mathbf{x}^*\mathbf{y}^*}(\mathbf{y}_t - \mathbf{y}^*) + o(\|\mathbf{w}_t - \mathbf{w}^*\|_2)$$
$$\nabla_{\mathbf{y}} f(\mathbf{x}_t, \mathbf{y}_t) = \mathbf{H}_{\mathbf{y}^*\mathbf{y}^*}(\mathbf{y}_t - \mathbf{y}^*) + \mathbf{H}_{\mathbf{y}^*\mathbf{x}^*}(\mathbf{x}_t - \mathbf{x}^*) + o(\|\mathbf{w}_t - \mathbf{w}^*\|_2).$$

Inserting the above formulas into the iteration scheme, it yields

$$\mathbf{w}_{t+1} = \begin{bmatrix} \mathbf{x}_{t+1} \\ \mathbf{y}_{t+1} \end{bmatrix} = \begin{bmatrix} \mathbf{I} - \eta \mathbf{H}_{\mathbf{x}^*\mathbf{x}^*} & -\eta \mathbf{H}_{\mathbf{x}^*\mathbf{y}^*} \\ \eta \mathbf{H}_{\mathbf{y}^*\mathbf{x}^*} & \mathbf{I} + \eta \mathbf{H}_{\mathbf{y}^*\mathbf{y}^*} \end{bmatrix} \begin{bmatrix} \mathbf{x}_t \\ \mathbf{y}_t \end{bmatrix} + \begin{bmatrix} \eta \mathbf{H}_{\mathbf{x}^*\mathbf{x}^*} \mathbf{x}^* + \eta \mathbf{H}_{\mathbf{x}^*\mathbf{y}^*} \mathbf{y}^* + \epsilon \\ -\eta \mathbf{H}_{\mathbf{y}^*\mathbf{y}^*} \mathbf{y}^* - \eta \mathbf{H}_{\mathbf{x}^*\mathbf{y}^*} \mathbf{x}^* + \epsilon \end{bmatrix}$$

where $\epsilon$ denotes the higher order error $o(\|\mathbf{w}_t - \mathbf{w}^*\|_2)$. According to Theorem 2.2, we know that simultaneous `GDA-AM` is equivalent to applying GMRES to solve the following linear system

$$(\mathbf{I} - \begin{bmatrix} (1-\alpha)\mathbf{I} - \eta \mathbf{H}_{\mathbf{x}^*\mathbf{x}^*} & -\eta \mathbf{H}_{\mathbf{x}^*\mathbf{y}^*} \\ \eta \mathbf{H}_{\mathbf{y}^*\mathbf{x}^*} & (1-\alpha)\mathbf{I} + \eta \mathbf{H}_{\mathbf{y}^*\mathbf{y}^*} \end{bmatrix})\mathbf{w} = \begin{bmatrix} \alpha \mathbf{I} + \eta \mathbf{H}_{\mathbf{x}^*\mathbf{x}^*} & \eta \mathbf{H}_{\mathbf{x}^*\mathbf{y}^*} \\ -\eta \mathbf{H}_{\mathbf{y}^*\mathbf{x}^*} & \alpha \mathbf{I} - \eta \mathbf{H}_{\mathbf{y}^*\mathbf{y}^*} \end{bmatrix} \mathbf{w} = \mathbf{b} + \epsilon$$

where $\mathbf{b} = \begin{bmatrix} \eta \mathbf{H}_{\mathbf{x}^*\mathbf{x}^*} \mathbf{x}^* + \eta \mathbf{H}_{\mathbf{x}^*\mathbf{y}^*} \mathbf{y}^* \\ -\eta \mathbf{H}_{\mathbf{y}^*\mathbf{y}^*} \mathbf{y}^* - \eta \mathbf{H}_{\mathbf{x}^*\mathbf{y}^*} \mathbf{x}^* \end{bmatrix}$. We now know that `GDA-AM` is equivalent to GMRES being applied to solve the following linear system

$$\begin{bmatrix} \alpha \mathbf{I} + \eta \mathbf{H}_{\mathbf{x}^*\mathbf{x}^*} & \eta \mathbf{H}_{\mathbf{x}^*\mathbf{y}^*} \\ -\eta \mathbf{H}_{\mathbf{y}^*\mathbf{x}^*} & \alpha \mathbf{I} - \eta \mathbf{H}_{\mathbf{y}^*\mathbf{y}^*} \end{bmatrix} \tilde{\mathbf{w}} = \mathbf{b}$$

The symmetric part of the coefficient matrix of the above linear system is

$$\begin{bmatrix} \alpha\mathbf{I} + \eta\mathbf{H}_{\mathbf{x}^*\mathbf{x}^*} & \mathbf{0} \\ \mathbf{0} & \alpha\mathbf{I} - \eta\mathbf{H}_{\mathbf{y}^*\mathbf{y}^*} \end{bmatrix}.$$

According to Proposition 2, $\alpha\mathbf{I} - \eta\mathbf{H}_{\mathbf{y}^*\mathbf{y}^*}$ is positive definite since $\mathbf{H}_{\mathbf{y}^*\mathbf{y}^*} \preccurlyeq \mathbf{0}$. If $\mathbf{H}_{\mathbf{x}^*\mathbf{x}^*}$ is positive semidefinite, then $\alpha\mathbf{I} + \eta\mathbf{H}_{\mathbf{x}^*\mathbf{x}^*}$ is positive definite and we're done. Otherwise, assume $\lambda_{\min}(\mathbf{H}_{\mathbf{x}^*\mathbf{x}^*}) < 0$. Then for fixed $\alpha$, when $\eta < -\frac{\alpha}{\lambda_{\min}(\mathbf{H}_{\mathbf{x}^*\mathbf{x}^*})}$, $\alpha\mathbf{I} + \eta\mathbf{H}_{\mathbf{x}^*\mathbf{x}^*}$ will be positive definite. Then according to Theorem 2.2, we know `GDA-AM` indeed converges. Let's create a new companion linear system as follows

$$\begin{bmatrix} \alpha\mathbf{I} + \eta\mathbf{H}_{\mathbf{x}^*\mathbf{x}^*} & \eta\mathbf{H}_{\mathbf{x}^*\mathbf{y}^*} \\ -\eta\mathbf{H}_{\mathbf{y}^*\mathbf{x}^*} & \alpha\mathbf{I} - \eta\mathbf{H}_{\mathbf{y}^*\mathbf{y}^*} \end{bmatrix} \hat{\mathbf{w}} = \mathbf{b} + \alpha\mathbf{w}^*$$

Note that $\hat{\mathbf{w}} = \mathbf{w}^*$ and GMRES on this companion linear system is convergent under suitable choice of learning rate $\eta$. Let the iterates of GMRES for $\tilde{\mathbf{w}}, \hat{\mathbf{w}}, \mathbf{w}$ be denoted by $\tilde{\mathbf{w}}_t, \hat{\mathbf{w}}_t, \mathbf{w}_t$. Then $\|\tilde{\mathbf{w}}_t - \hat{\mathbf{w}}_t\| \leq \|\tilde{\mathbf{w}}_t - \mathbf{w}_t\| + \|\hat{\mathbf{w}}_t - \mathbf{w}_t\|$. According to Theorem C.5, we also have $\|\tilde{\mathbf{w}}_t - \mathbf{w}_t\| \leq \sigma_k\epsilon, 1 \leq k \leq t$. Further more, again according to Theorem C.5, we know $\|\hat{\mathbf{w}}_t - \hat{\mathbf{w}}_t\| \leq \sigma_k(\alpha\mathbf{w}^* + \epsilon)$. Starting from an initial point very close to $\mathbf{w}^*$ and let $t \to \infty$ and $\alpha, \epsilon \to 0$, $\hat{\mathbf{w}}_t$ will converge to $\mathbf{w}^* = (\mathbf{x}^*, \mathbf{y}^*)$, which means the local minimax $\mathbf{w}^* = (\mathbf{x}^*, \mathbf{y}^*)$ is a limiting point of GDA-RAM. $\quad\square$

**Theorem C.6.** *For strongly-convex-strongly-concave function $f(\mathbf{x}, \mathbf{y})$, `GDA-AM` will converge to the Nash equilibrium of this function.*

***Proof:*** Since strongly-convex-strongly-concave function $f(\mathbf{x}, \mathbf{y})$ has unique Nash equilibrium which is also the unique minimax point, this minimax point must be the limiting point of `GDA-AM` according to Theorem C.4. $\quad\square$

### C.4.1 BILINEAR-QUADRATIC GAMES

Moreover, we can further show that the `GDA-AM` converges on bilinear-quadratic games. Consider a quadratic problem as follows,

$$\min_{\mathbf{x}\in\mathbb{R}^n} \max_{\mathbf{y}\in\mathbb{R}^n} f(\mathbf{x}, \mathbf{y}) = \mathbf{x}^T\mathbf{A}\mathbf{y} + \mathbf{x}^T\mathbf{B}\mathbf{x} - \mathbf{y}^T\mathbf{C}\mathbf{y} + \mathbf{b}^T\mathbf{x} + \mathbf{c}^T\mathbf{y}, \tag{36}$$

where $\mathbf{A}$ is full rank, $\mathbf{B}$ and $\mathbf{C}$ are both positive definite.

**Theorem C.7.** *[Global convergence for simultaneous `GDA-AM` on bilinear-quadratic problem] Let $\mathbf{r}_t^{(Sim)}$ be the residual of Algorithm 2 being applied to problem equation 36. For some constant $\rho < 1$,*

$$\|\mathbf{r}_t^{(Sim)}\|_2 \leq \underbrace{\left(1 - \frac{(\lambda_{\min}(\mathbf{J}^T + \mathbf{J}))^2}{4\lambda_{\max}(\mathbf{J}^T\mathbf{J})}\right)^{t/2}}_{\rho^{t/2}} \|\mathbf{r}_0\|_2, \tag{37}$$

*where $\mathbf{J} = \begin{bmatrix} \eta\mathbf{B} & \eta\mathbf{A} \\ -\eta\mathbf{A}^T & \eta\mathbf{C} \end{bmatrix}$ and $\lambda_{\min}$ and $\lambda_{\max}$ denote the smallest and largest eigenvalue, respectively.*

The convergence property of GMRES has been studied in the next theorem. We use this theorem to show the convergence rate of `GDA-AM` for bilinear-quadratic games.

**Theorem C.8** (Elman (1982)). *Consider solving a linear system $\mathbf{E}\mathbf{x} = \mathbf{b}$ using GMRES. Let $\mathbf{r}_t = \mathbf{b} - \mathbf{E}\mathbf{x}_t$ be the residual at $t$th iteration. If the Hermitian part of $\mathbf{E}$ is positive definite, then for some positive constant $\rho < 1$, it holds that*

$$\|\mathbf{r}_t\|_2 \leq \underbrace{\left(1 - \frac{(\lambda_{\min}(\mathbf{E}^H + \mathbf{E}))^2}{4\lambda_{\max}(\mathbf{E}^H\mathbf{E})}\right)^{t/2}}_{\rho^{t/2}} \|\mathbf{r}_0\|_2. \tag{38}$$

***Proof of Theorem C.7.*** Applying simultaneous `GDA-AM` to solve the above problem is equivalent to applying Anderson Mixing on the following fixed point iteration:

$$
\begin{bmatrix} \mathbf{x}_{t+1} \\ \mathbf{y}_{t+1} \end{bmatrix} = \underbrace{\begin{bmatrix} \mathbf{I} - \eta\mathbf{B} & -\eta\mathbf{A} \\ \eta\mathbf{A}^T & \mathbf{I} - \eta\mathbf{C} \end{bmatrix}}_{\mathbf{G}^{(Quad-sim)}} \underbrace{\begin{bmatrix} \mathbf{x}_t \\ \mathbf{y}_t \end{bmatrix}}_{\mathbf{w}_t^{(Quad-sim)}} + \underbrace{\begin{bmatrix} -\eta\mathbf{b} \\ -\eta\mathbf{c} \end{bmatrix}}_{\mathbf{b}^{(Quad-sim)}} .
\tag{39}
$$

We know that we need to study the convergence properties of GMRES for solving the following linear system

$$
\begin{bmatrix} \eta\mathbf{B} & \eta\mathbf{A} \\ -\eta\mathbf{A}^T & \eta\mathbf{C} \end{bmatrix} \mathbf{w} = \mathbf{b}.
\tag{40}
$$

For notational simplicity, the superscripts has been dropped. Denote the coefficient matrix $\begin{bmatrix} \eta\mathbf{B} & \eta\mathbf{A} \\ -\eta\mathbf{A}^T & \eta\mathbf{C} \end{bmatrix}$ by $\mathbf{J}$. The symmetric part of $\mathbf{J}$ is

$$
\frac{\mathbf{J} + \mathbf{J}^T}{2} = \begin{bmatrix} \frac{\eta}{2}(\mathbf{B} + \mathbf{B}^T) & \mathbf{0} \\ \mathbf{0} & \frac{\eta}{2}(\mathbf{C} + \mathbf{C}^T) \end{bmatrix}
$$

which is positive definite. Then immediately by Theorem C.8, the following convergence rate holds For some constant $0 < \rho < 1$,

$$
\|\mathbf{r}_t\|_2 = \min_{p \in \mathbb{P}_t^1} \|p(\mathbf{J})\mathbf{r}_0\|_2 \leq \underbrace{\left( 1 - \frac{(\lambda_{\min}(\mathbf{J} + \mathbf{J}^T)^2}{(4\lambda_{\max}(\mathbf{J}^T\mathbf{J}))} \right)^{t/2}}_{\rho^{t/2}} \|\mathbf{r}_0\|_2
\tag{41}
$$

$$
= \rho^{t/2} \|\mathbf{r}_0\|_2
$$

$\square$

Note that the convergence of `GDA-AM` for bilinear-quadratic games can also be analyzed by numerical range as shown in (Bollapragada et al., 2018). Although we previously show that analysis based on the numerical range can not help us derive a convergent bound for bilinear games, we show analysis in Bollapragada et al. (2018) can be extended to bilinear-quadratic games. When $\mathbf{B}$ and $\mathbf{C}$ are positive definite, 1 is outside of the numerical range of matrix $\mathbf{G}^{(Quad-sim)}$ as shown in 7a. When $\mathbf{B}$ or $\mathbf{C}$ is not positive definite, 1 can be included in the numerical range of matrix $\mathbf{G}^{(Quad-sim)}$ as shown in 7b. That is saying analysis based on the numerical range (Crouzeix & Palencia, 2017; Bollapragada et al., 2018) to the bilinear-quadratic problem can lead to a convergent result when $\mathbf{B}$ and $\mathbf{C}$ are positive definite. And analysis based on the numerical range can not help us derive convergent results when $\mathbf{B}$ or $\mathbf{C}$ is not positive definite.

## C.5 STOCHASTIC CONVEX-NONCONVACE CASE

In this section, we study the convergence of `GDA-AM` for convex-noncovace problem in the stochastic setting with the same assumptions in Wei et al. (2021b); Xu et al. (2021). The recent work Wei et al. (2021b) proves the convergence of the stochastic gradient descent with Anderson Mixing for min optimization. The convergence of `GDA-AM` for minimax optimization builds on top of it with several modifications. The minimax problem is equivalent to minimizing a function $\Phi(\cdot) = \max_{\mathbf{y} \in \mathcal{Y}} f(\cdot, \mathbf{y})$ (Lin et al., 2020). And we are interested in complexity of a pair of $\epsilon$-stationary point $(x, y)$ instead of analysis of a point $x$.

**Definition 5.** *(Lin et al., 2020) A pair of points $(\mathbf{x}, \mathbf{y})$ is an $\epsilon$-stationary point $(\epsilon \geq 0)$ of a differentiable function $\Phi$ if*

$$
\|\nabla_{\mathbf{x}} f(\mathbf{x}, \mathbf{y})\| \leq \epsilon
$$
$$
\|\mathcal{P}_{\mathcal{Y}} (\mathbf{y} + (1/\ell)\nabla_{\mathbf{y}} f(\mathbf{x}, \mathbf{y})) - \mathbf{y}\| \leq \epsilon/\ell
$$

**Assumption 1.** *$f : \mathbb{R}^d \mapsto \mathbb{R}$ is continuously differentiable. $f(x) \geq f^{low} > -\infty$ for any $x \in \mathbb{R}^d$. $\nabla f$ is globally L-Lipschitz continuous; namely $\|\nabla f(x) - \nabla f(y)\|_2 \leq L\|x - y\|_2$ for any $x, y \in \mathbb{R}^d$.*

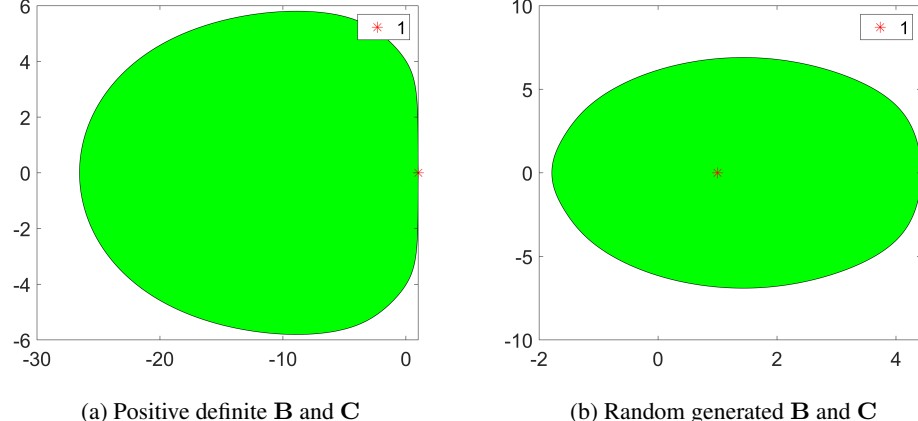

(a) Positive definite $\mathbf{B}$ and $\mathbf{C}$        (b) Random generated $\mathbf{B}$ and $\mathbf{C}$

Figure 7: Numerical range of fixed-point operator (Simultaneous `GDA-AM` ) $\mathbf{G} = \begin{bmatrix} \mathbf{I} - \eta\mathbf{B} & -\eta\mathbf{A} \\ \eta\mathbf{A}^T & I - \eta\mathbf{C} \end{bmatrix}$ for bilinear-quadratic games.

**Assumption 2.** *For any iteration $k$, the stochastic gradient $\nabla f_{\xi_k}(x_k)$ satisfies $\mathbb{E}_{\xi_k}\left[\nabla f_{\xi_k}(x_k)\right] = \nabla f(x_k)$, $\mathbb{E}_{\xi_k}\left[\|\nabla f_{\xi_k}(x_k) - \nabla f(x_k)\|_2^2\right] \leq \sigma^2$, where $\sigma > 0$, and $\xi_k, k = 0, 1, \ldots$, are independent samples that are independent of $\{x_i\}^k$*

**Theorem C.9.** *For a general convex-nonconcave function $f$, suppose that Assumptions 1 and 2 hold. Batch size $n_t = n$ for $t = 0, \ldots, N-1$. $C > 0$ is a constant. $\beta_t = \frac{\mu}{4L(1+C^{-1})} \cdot \delta_t \geq C\beta_t^{-2}, 0 \leq \alpha_t \leq \min\left\{1, \beta_t^{\frac{1}{2}}\right\}$ and $\alpha_t$ is chosen to make sure the positive definiteness of $H_t$. Let $R$ be a random variable following $P_R(t) \stackrel{def}{=} \mathrm{Prob}\{R = t\} = 1/N$, and $\bar{N}$ be the total number of stochastic GDA-AM calls needed to calculate stochastic gradients $\tilde{\nabla} f_{S_t}(\mathbf{w}_t)$ in our algorithm. To ensure $\mathbb{E}\left[\left\|\tilde{\nabla} f(\mathbf{w}_R)\right\|_2\right] \leq \epsilon$, total number of stochastic GDA-AM calls needed to calculate stochastic gradients $\tilde{\nabla} f_{S_t}(\mathbf{w}_t)$ is $O(\epsilon^{-4})$.*

Recall that we can recast GDA scheme as the following fixed point iteration.

$$\mathbf{w}_{t+1} = G_\eta^{(\text{sim})}(\mathbf{w}_t) \triangleq \mathbf{w}_t + \eta V(\mathbf{w}_t) \text{ with } \mathbf{w} = \begin{bmatrix} \mathbf{x} \\ \mathbf{y} \end{bmatrix}, V(\mathbf{w}) = \begin{bmatrix} -\nabla_{\mathbf{x}} f(\mathbf{x}, \mathbf{y}) \\ \nabla_{\mathbf{y}} f(\mathbf{x}, \mathbf{y}) \end{bmatrix}$$

Ignoring the stepsize $\eta$ and let $\mathbf{W}_t$ and $\mathbf{R}_t$ record the first and second order diffrence of recent m iterates:

$$\mathbf{W}_t = [\Delta\mathbf{w}_{t-m}, \Delta\mathbf{w}_{t-m+1}, \cdots, \Delta\mathbf{w}_{t-1}], \mathbf{R}_t = [\Delta V_{t-m}, \Delta V_{t-m+1}, \cdots, \Delta V_{t-1}]$$

Similarly as Wei et al. (2021b),the Anderson mixing can be decoupled into

$$\bar{\mathbf{w}}_{t+1} = \mathbf{w}_t - \mathbf{W}_t \Gamma_t, \quad \text{(Projection step)}$$
$$\bar{\mathbf{w}}_{t+1} = \mathbf{w}_t + \beta_t \bar{V}_t, \quad \text{(Mixing step)}$$

where $\beta_t$ is the mixing parameter, and $\bar{V}_t = V_t - \mathbf{W}_t \Gamma_t$ and $\Gamma_t$ is solved by

$$\Gamma_t = \underset{\Gamma \in \mathbb{R}^m}{\arg\min} \|V_t - \mathbf{R_t}\Gamma\|_2 + \delta_t \|\Gamma\|_2$$

We want to argue that similar arguments in Wei et al. (2021b) can be applied to the problem here. To see why Anderson mixing works for minimax optimization, we assume function $f$ is smooth. Then the hessian matrix for $G_\eta^{(\text{sim})}$ is

$$H = \begin{pmatrix} -\nabla_{\mathbf{xx}}^2 f & -\nabla_{\mathbf{xy}}^2 f \\ \nabla_{\mathbf{yx}}^2 f & \nabla_{\mathbf{yy}}^2 f \end{pmatrix}$$

Notice that in a small neighborhood of $\mathbf{w}_{t+1}$, we have

$$\mathbf{R_t} = -H\mathbf{W}_t = \begin{pmatrix} \nabla^2_{\mathbf{xx}} f & \nabla^2_{\mathbf{xy}} f \\ -\nabla^2_{\mathbf{yx}} f & -\nabla^2_{\mathbf{yy}} f \end{pmatrix} \mathbf{W}_t$$

Thus $\|V_t - \mathbf{R_t}\Gamma\|_2 \approx \|V_t + H\mathbf{W}_t\Gamma\|_2$, which is equivalent to solving for a vector $p_t$ such that $Hp_k = V_t$. This is exactly the second order method for the fixed point iteration problem. Also at each step the AM is minimizing the residual, the reason that AM is equivalent to GMRES for linear problem is that this quadratic approximation is exact. Finally, we rewrite AM as the quasi-newton framework as Wei et al. (2021b) did. $\mathbf{w}_{t+1} = \mathbf{w}_t + H_t V_t$ where

$$\min_{H_t} \|H_t - \beta_t I\|_F \text{ subject to } H_t \mathbf{R}_t = -X_t$$

Finally, with damping parameter, Anderson mixing has the following form

$$\mathbf{W}_{t+1} = \mathbf{W}_t + \beta_t V_t - \alpha_t \left( \mathbf{W}_t + \beta_t \mathbf{R_t} \right) \Gamma_t \tag{42}$$

we can also apply the very similar arguments to prove key results in lemma 1, lemma 2 in Wei et al. (2021b). There is also a key difference with Wei et al. (2021b). Here we are considering minimax optimization problem. Thus our gradient is actually $V(\mathbf{w}) = \begin{bmatrix} -\nabla_{\mathbf{x}} f(\mathbf{x}, \mathbf{y}) \\ \nabla_{\mathbf{y}} f(\mathbf{x}, \mathbf{y}) \end{bmatrix}$ rather than $\nabla f(\mathbf{w}) = \begin{bmatrix} \nabla_{\mathbf{x}} f(\mathbf{x}, \mathbf{y}) \\ \nabla_{\mathbf{y}} f(\mathbf{x}, \mathbf{y}) \end{bmatrix}$ This will introduce some difficulty to the dynamics of the fixed pointe iteration. However, noticing that $\|V\| = \|\nabla f(\mathbf{w})\|$ and

$$f(\mathbf{w}_{t+1}) \leq f(\mathbf{w}_t) + \nabla f(\mathbf{w}_t)^{\mathrm{T}} (\mathbf{w}_{t+1} - \mathbf{w}_t) + \frac{L}{2} \|\mathbf{w}_{t+1} - \mathbf{w}_t\|_2^2$$

$$\leq f(\mathbf{w}_t) + \tilde{\nabla} f(\mathbf{w}_t)^{\mathrm{T}} (\mathbf{w}_{t+1} - \mathbf{w}_t) + \frac{L}{2} \|\mathbf{w}_{t+1} - \mathbf{w}_t\|_2^2 \tag{43}$$

$$= f(\mathbf{w}_t) + \tilde{\nabla} f(\mathbf{w}_t)^{\mathrm{T}} H_t V_t + \frac{L}{2} \|H_t V_t\|_2^2$$

where

$$\tilde{\nabla} f(\mathbf{w}_t) = \begin{bmatrix} -\nabla_{\mathbf{x}} f(\mathbf{x}, \mathbf{y}) \\ \nabla_{\mathbf{y}} f(\mathbf{x}, \mathbf{y}) \end{bmatrix} \tag{44}$$

we call this the ascent-descent gradient (ADG) which is the gradient for minimax optimization problem

$$\min_{\mathbf{x} \in \mathbb{R}^d} \max_{\mathbf{y} \in \mathbb{R}^d} f(\mathbf{x}, \mathbf{y}).$$

To see why $\nabla f(\mathbf{w}_t)^{\mathrm{T}} (\mathbf{w}_{t+1} - \mathbf{w}_t) \leq \tilde{\nabla} f(\mathbf{w}_t)^{\mathrm{T}} (\mathbf{w}_{t+1} - \mathbf{w}_t)$, we consider their difference

$$(\tilde{\nabla} - \nabla) f(\mathbf{w}_t)^{\mathrm{T}} (\mathbf{w}_{t+1} - \mathbf{w}_t) = -2\nabla_{\mathbf{x}} f(\mathbf{x}_t, \mathbf{y}_t)^{T} (\mathbf{x}_{t+1} - \mathbf{x}_t).$$

For fixed $\mathbf{y}_t$, $f(\mathbf{x}_t, \mathbf{y}_{t+1})$ has the Talyor expansion:

$$f(\mathbf{x}_{t+1}, \mathbf{y}_t) = f(\mathbf{x}_t, \mathbf{y}_t) + \nabla_{\mathbf{x}} f(\mathbf{x}_t, \mathbf{y}_t)^{T} (\mathbf{x}_{t+1} - \mathbf{x}_t) + (\mathbf{x}_{t+1} - \mathbf{x}_t)^{T} \nabla_{\mathbf{xx}} f(\mathbf{x}_t + \theta(\mathbf{x}_{t+1} - \mathbf{x}_t), \mathbf{y}_t)(\mathbf{x}_{t+1} - \mathbf{x}_t)$$

Assuming f is convex w.r.t $\mathbf{x}$ and apply safeguard to ensure $f(\mathbf{x}_{t+1}, \mathbf{y}_t) \leq f(\mathbf{x}_t, \mathbf{y}_t)$ can guarantee $(\tilde{\nabla} - \nabla) f(\mathbf{w}_t)^{\mathrm{T}} (\mathbf{w}_{t+1} - \mathbf{w}_t) \geq 0$. Now applying lemmas in Wei et al. (2021b), we can derive the convergence of our method for general convex-nonconcave function similarly.

# D  ADDITIONAL EXPERIMENTS

## D.1  COMPARISON WITH EG WITH POSITIVE MOMENTUM

In this section, we include additional comparison between `GDA-AM` and EG with positive momentum. `GDA-AM` has two big theoretical advantages over EG with positive momentum. First, convergence of `GDA-AM` does not require strong assumptions on choices of hyperparamters. Second, 4.1 and 4.2 provide nonasymptotic guarantees while convergence of EG with positive mometum is asymptotic. Experimental results are shown in 8. It indicates `GDA-AM` outperforms EG with positive momentum. Finding a good choice of the inner and outer step size of EG and momentum term is hard. For EG with positive momentum, we set the step size of extrapolation step as 1, the step size of update as 0.5, and the positive momentum term as 0.3 after grid search as shown in 8b and 8c. On the other hand, `GDA-AM` converges fast for different step size without hyperparameter tuning.

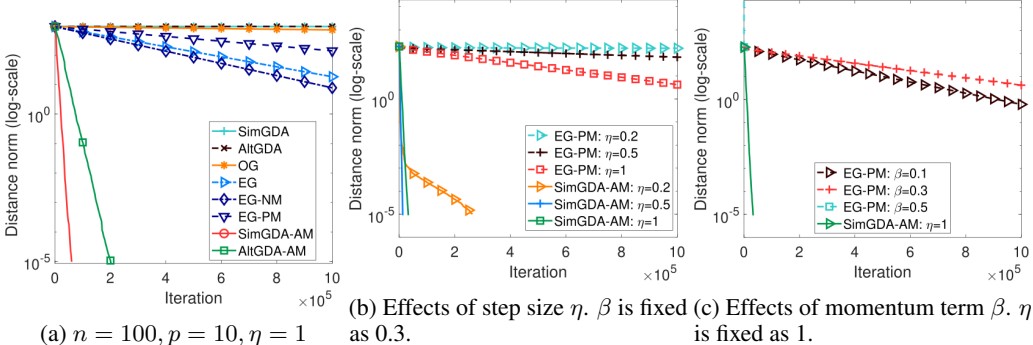

(a) $n = 100, p = 10, \eta = 1$  (b) Effects of step size $\eta$. $\beta$ is fixed as 0.3.  (c) Effects of momentum term $\beta$. $\eta$ is fixed as 1.

Figure 8: Additional Comparison between `GDA-AM` and EG with positive momentum

## D.2    1D MINIMAX FUNCTIONS

We begin with investigating the empirical performance of `GDA-AM` for 6 non-trivial 1d bivariate functions. We set initial points as $(3, 3)$ and $m$ as 20 or 5 for all functions. We use optimal learning rates for all methods on each problem. Results are shown in Figure 9, 10, 11, 12, 13 and 14. We observe `GDA-AM` consistently outperforms all baselines and improves convergence. It is worthwhile to mention that the difference between `GDA-AM` and traditional averaging is twofold. First, traditional averaging does not involve an adaptive averaging scheme and thus blindly converge to $(0, 0)$ for all 1d bivariate functions. In contrast, `GDA-AM` obtains optimal weights by solving a small linear system on past iterates. Using different weights for each iteration, `GDA-AM` is able to minimize the residual of past iterates and thus find the solution of a fixed-point iteration. More importantly, averaging does not change the GDA dynamic because averaging generates a new sequence of parameters based on GDA iterates. This means averaging is independent with base training algorithm (GDA here). However, `GDA-AM` changes the dynamic directly by overwriting the latest iterate. It means Anderson Mixing interacts with GDA, which is another major difference from averaging.

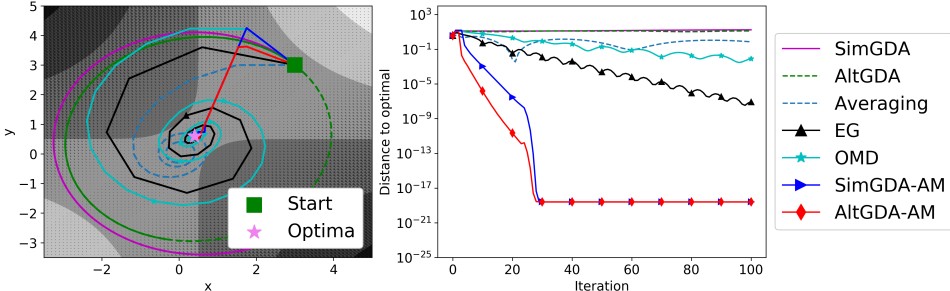

Figure 9: $f(x, y) = (x - \frac{1}{2})(y - \frac{1}{2}) + \frac{1}{3}e^{-(x-0.25)^2 - (y-0.75)^2}$. The optima for this function is not $(0, 0)$. Because averaging blindly converges to $(0, 0)$, it can never find the correct solution.

## D.3    DENSITY ESTIMATION

To test our proposed method, we evaluate our method on two low-dimension density estimation problems, mixture of 25 Gaussians and Swiss roll. For both generator and discriminator, we use fully connected neural networks with 3 hidden layers and 128 hidden units in each layer. Except for the output layer of discriminator that uses a sigmoid activation, we use tanh-activation for all other layers. We run Adam and `GDA-AM` for 50000 steps. The learning rate is set as $2 \times 10^{-4}$ and $\beta_1 = 0, \beta_2 = 0.9$ after an extensive grid search, which is close to the maximal possible stepsize under which the methods rarely diverge. Figure 15 and 16 show the output after $\{1K, 10K, 30K, 50K\}$ iterations. It can be seen that our method converges faster to the target distribution offers a improvement over Adam. In addition, we can observe that the generated samples using our method gather around the circle and are less connected with other circles.

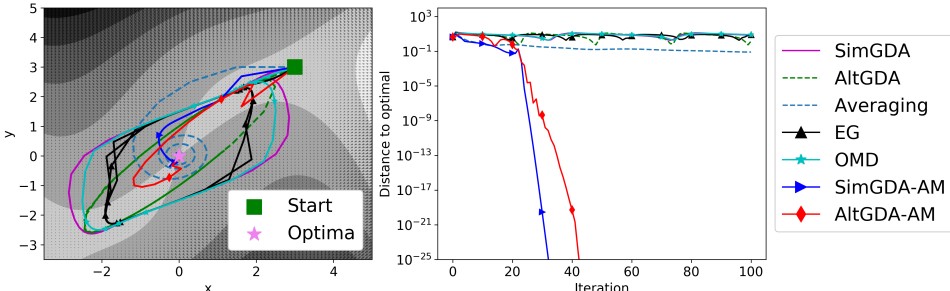

Figure 10: $f(x, y) = (4x^2 - (y - 3x + 0.05x^3)^2 - 0.1y^4)e^{-0.01(x^2+y^2)}$. All baselines except averaging are cyclying around the optima. Averaging is converging slowly.

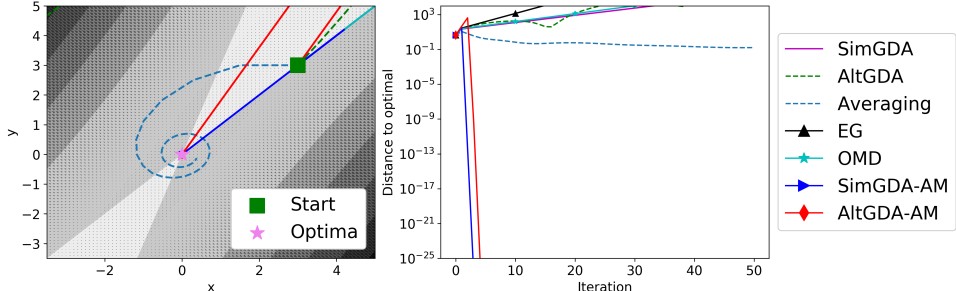

Figure 11: $f(x, y) = -3x^2 - y^2 + 4xy$. Baselines tend to diverge. Averaging is converging slowly again because averaging can only blindly converge to $(0, 0)$ and the optima for this function is $(0, 0)$.

### D.4 ROBUST NEURAL NETWORK TRAINING

In this section, we test the effectiveness of `GDA-AM` by training a robust neural network on MNIST data set against adversarial attacks (Madry et al., 2019; Goodfellow et al., 2015; Kurakin et al., 2017) . The optimization formulation is

$$\min_{\mathbf{w}} \sum_{i=1}^{N} \max_{\delta_i, \text{ s.t. } |\delta_i|_\infty \leq \varepsilon} \ell\left(f\left(x_i + \delta_i; \mathbf{w}\right), y_i\right) \tag{45}$$

where $w$ is the parameter of the neural network, the pair $(x_i, y_i)$ denotes the $i$-th data point, and $\delta_i$ is the perturbation added to data point $i$. The accuracy of our formulation against popular attacks, FGSM (Goodfellow et al., 2015) and PGD (Kurakin et al., 2017), are summarized in Table 2.. Since solving such problem is computationally challenging, Nouiehed et al. (2019) proposed an approximation of the above optimization problem with a new objective function as the following nonconvex-concave problem:

$$\min_{\mathbf{w}} \sum_{i=1}^{N} \max_{\mathbf{t} \in \mathcal{T}} \sum_{j=0}^{9} t_j \ell\left(f\left(x_{ij}^K; \mathbf{w}\right), y_i\right), \mathcal{T} = \left\{(t_1, \cdots, t_m) \mid \sum_{i=1}^{m} t_i = 1, t_i \geq 0\right\} \tag{46}$$

where $K$ is a parameter in the approximation, and $x_{ij}^K$ is an approximated attack on sample $x_i$ by changing the output of the network to label $j$. We use the public available implementation (Nouiehed et al., 2019) [2]. We apply our algorithm on top of (Nouiehed et al., 2019) and compare our results $(p = 50)$ with (Madry et al., 2019; Zhang et al., 2019; 2020; Nouiehed et al., 2019). Results are summarized in table 2. We can observe that `GDA-AM` leads to a comparable or slightly better performance to the other methods. In addition, `GDA-AM` does not exhibit a significant drop in accuracy when $\epsilon$ is larger and this suggests the learned model is more robust.

---

[2]https://github.com/optimization-for-data-driven-science/Robust-NN-Training

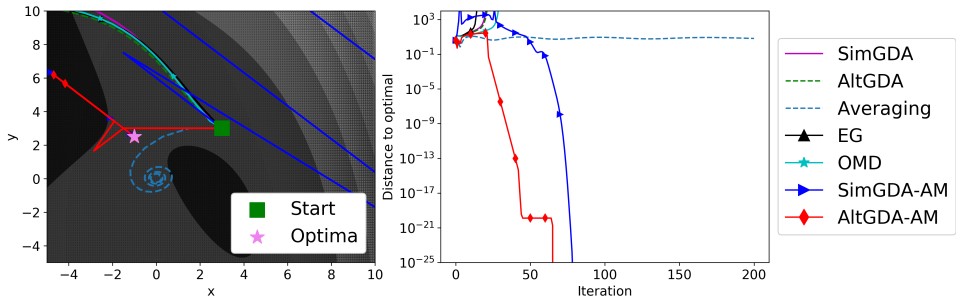

Figure 12: $f(x, y) = \frac{1}{3}x^3 + y^2 + 2xy - 6x - 3y + 4$.

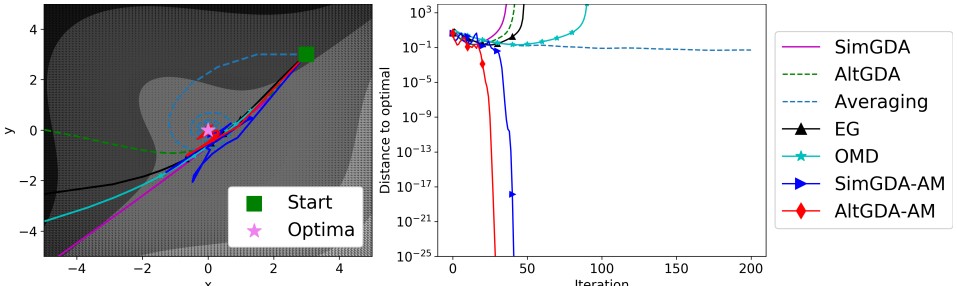

Figure 13: $f(x, y) = x^3 - y^3 - 2xy + 6$.

### D.5 IMAGE GENERATION

In this section, we provide additional experimental results that are not given in Section 5. Figure 18a and 18b show the Inception Score for CIFAR10 using WGAN-GP and SNGAN. It can be observed that our method consistently performs better than Adam and EG during training. Further, on CIFAR-10 using WGAN-GP and SNGAN, GDA-AM is slightly slower than Adam (about 110-115% computational time), but significantly faster than EG (about 65-75% computational time).

### D.6 DETAILS ON THE EXPERIMENTS

For our experiments, we used the PyTorch [3] deep learning framework. Experiments were run one NVIDIA V100 GPU. The residual network architecture for generator and discriminator are summarized in Table 3 and 4. We use a WGAN-GP loss, with gradient penalty $\lambda = 10$. When using the gradient penalty (WGAN-GP), we remove the batch normalization layers in the discriminator. When using SNGAN, we replace the batch normalization layers with spectral normalization. Hyperparamters of Adam are selected after grid search. We use a learning rate of $2 \times 10^{-4}$ and batch size of 64. For table size of GDA-AM , we set it as 120 for CIFAR10 and 150 for CelebA. We set $\beta_1 = 0.0$ and $\beta_2 = 0.9$ as we find it gives us better models than default settings.

---

[3]https://pytorch.org/

| | Natural | FGSM $L_\infty$ | | | PGD$^{40} L_\infty$ | | |
|---|---|---|---|---|---|---|---|
| | | $\varepsilon = 0.2$ | $\varepsilon = 0.3$ | $\varepsilon = 0.4$ | $\varepsilon = 0.2$ | $\varepsilon = 0.3$ | $\varepsilon = 0.4$ |
| Madry et al. (2019) | 98.58% | 96.09% | 94.82% | 89.84% | 94.64% | 91.41% | 78.67% |
| Trade: $\varepsilon = 0.35$ | 97.37% | 95.47% | 94.86% | 79.04% | 94.41% | 92.69% | 85.74% |
| Trade: $\varepsilon = 0.40$ | 97.21% | 96.19% | 96.17% | 96.14% | 95.01% | 94.36% | 94.11% |
| Nouiehed et al. (2019) | 98.20% | 97.04% | 96.66% | 96.23% | 96.00% | 95.17% | 94.22% |
| Zhang et al. (2020) | **98.89%** | **97.87%** | 97.23% | 95.81% | **96.71%** | 95.62% | 94.51% |
| GDA-AM | 98.61% | 97.75% | **97.74%** | **97.75%** | 96.47% | **95.91%** | **95.41%** |

Table 2: Test accuracies under FGSM and PGD attack. Trade refers to Zhang et al. (2019).

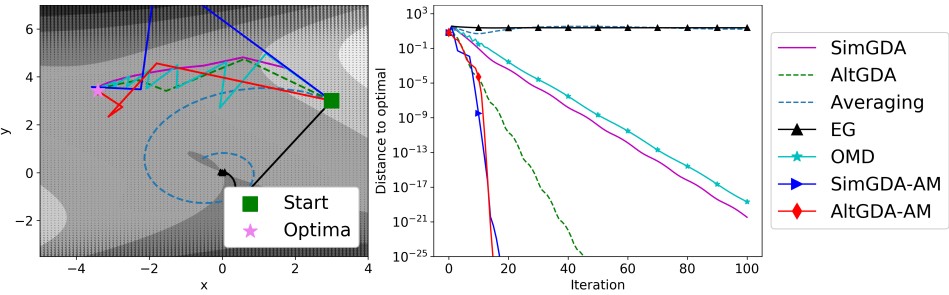

Figure 14: $f(x, y) = 2x^2 + y^2 + 4xy + \frac{4}{3}y^3 - \frac{1}{4}y^4$.

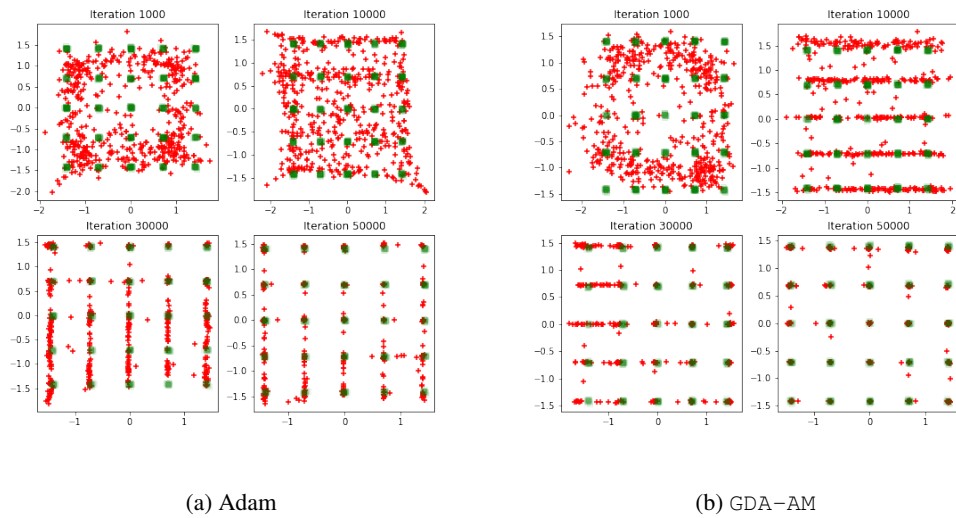

(a) Adam

(b) GDA-AM

Figure 15: **25 Gaussians**: Evolution plot of Adam and GDA-AM. Green dots are observed points and red dots are generated points.

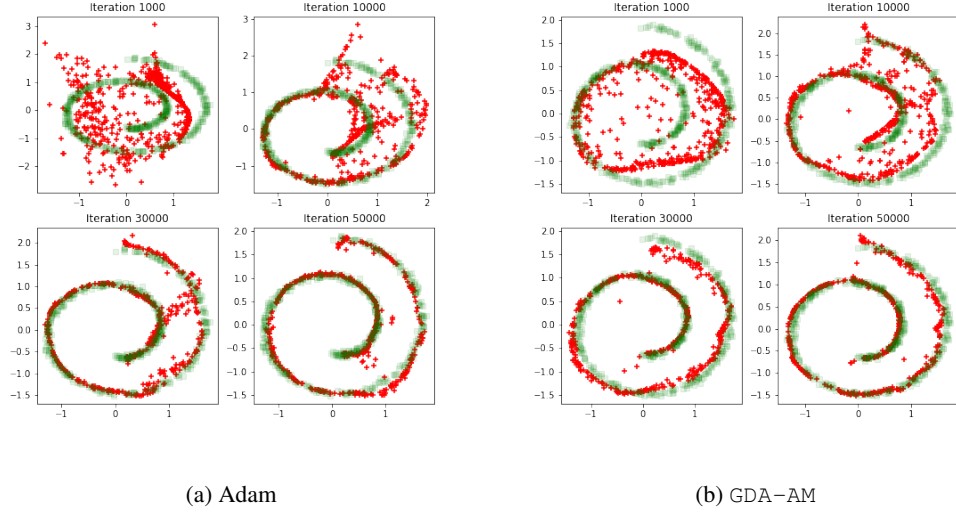

(a) Adam

(b) GDA-AM

Figure 16: **Swiss roll**: Evolution plot of Adam and GDA-AM. Green dots are observed points and red dots are generated points.

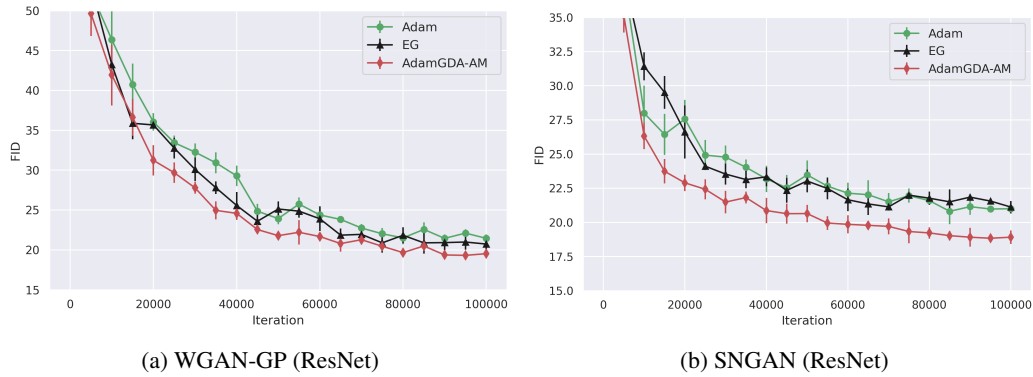

(a) WGAN-GP (ResNet)        (b) SNGAN (ResNet)

Figure 17: FID (lower or ↓ is better) for CIFAR 10

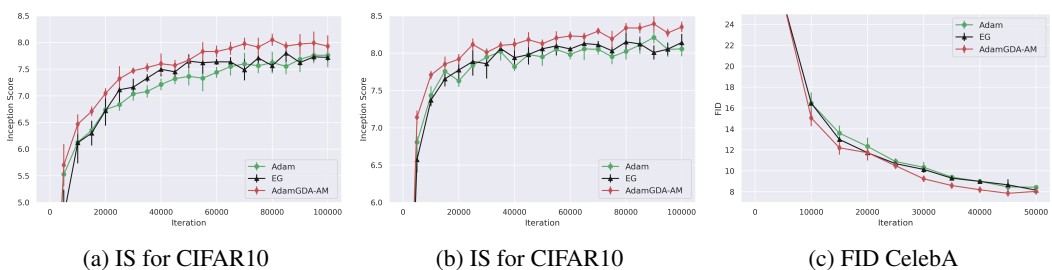

(a) IS for CIFAR10     (b) IS for CIFAR10     (c) FID CelebA

Figure 18: **Left:** IS for CIFAR10 using WGANGP. **Middle:** IS for CIFAR10 using SNGAN. **Right:** FID for CelebA using WGANGP.

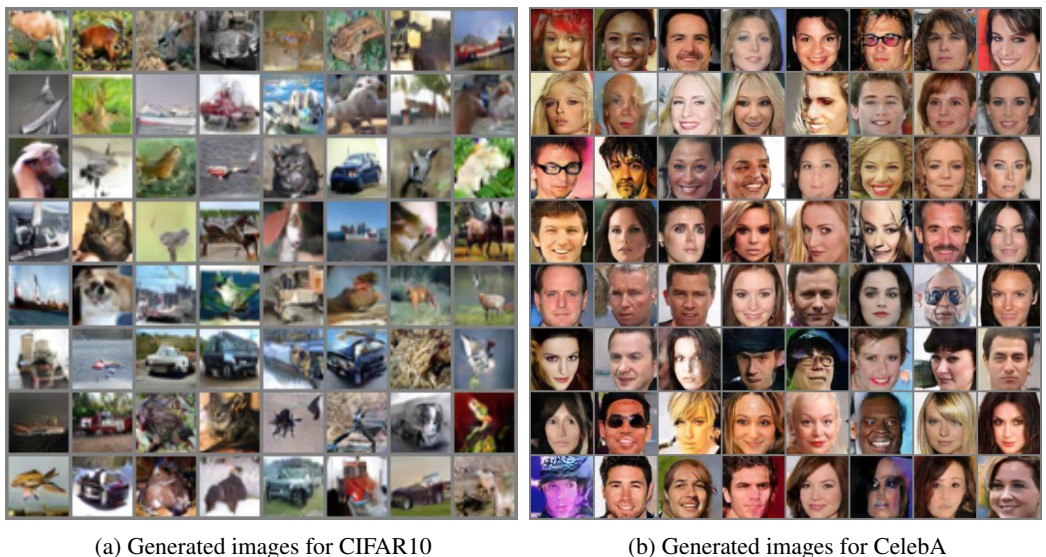

(a) Generated images for CIFAR10     (b) Generated images for CelebA

Figure 19: Generated Images for CIFAR10 and CelebA using WGAN-GP(ResNet)

Table 3: ResNet architecture used for our CIFAR-10 experiments.

| Generator |
| --- |
| Input: $z \in \mathbb{R}^{128} \sim \mathcal{N}(0, I)$ |
| Linear $128 \to 256 \times 4 \times 4$ |
| ResBlock $128 \to 128$ |
| ResBlock $256 \to 256$ |
| ResBlock $256 \to 256$ |
| Batch Normalization |
| ReLu |
| transposed conv. (256, kernel:$3 \times 3$, stride:1, pad: 1 |
| $\tanh(\cdot)$ |

| Discriminator |
| --- |
| Input: $x \in \mathbb{R}^{3 \times 32 \times 32}$ |
| Linear $128 \to 128 \times 4 \times 4$ |
| ResBlock $128 \to 128$ |
| ResBlock $128 \to 128$ |
| ResBlock $128 \to 128$ |
| Linear $128 \to 1$ |

Table 4: ResNet architecture used for our CelebA ($64 \times 64$) experiments.

| Generator |
| --- |
| Input: $z \in \mathbb{R}^{128} \sim \mathcal{N}(0, I)$ |
| Linear $128 \to 512 \times 8 \times 8$ |
| ResBlock $512 \to 256$ |
| ResBlock $256 \to 128$ |
| ResBlock $128 \to 64$ |
| Batch Normalization |
| ReLu |
| transposed conv. (64, kernel:$3 \times 3$, stride:1, pad: 1 |
| $\tanh(\cdot)$ |

| Discriminator |
| --- |
| Input: $x \in \mathbb{R}^{3 \times 64 \times 64}$ |
| Linear $128 \to 128 \times 4 \times 4$ |
| ResBlock $128 \to 128$ |
| ResBlock $128 \to 256$ |
| ResBlock $256 \to 512$ |
| Linear $512 \to 1$ |

