# OpenReview forum: "GDA-AM: ON THE EFFECTIVENESS OF SOLVING MIN-IMAX OPTIMIZATION VIA ANDERSON MIXING"
_ICLR.cc/2022/Conference — ICLR 2022 Poster_

### Official Review · Reviewer_KCYs · 2021-11-01

**Correctness:** 3
**Technical Novelty And Significance:** 1
**Empirical Novelty And Significance:** 3
**Recommendation:** 6
**Confidence:** 5

**Main Review:**

*** Post rebuttal ***

After a discussion with the authors, I updated my score to accept.

******


The biggest concern of the paper is the novelty. The authors present the Anderson mixing technique and its link with GMRES, up to page 4. The claimed novelty begins at section 3: GDA-AM: GDA WITH ANDERSON MIXING. The author claim that  their work "is the first work to improve the GDA dynamics by tapping into advanced fixed-point algorithms," although this idea was explicitly mentioned in a paper they cited:

> ---
> Recent work in optimization analysed adaptive algorithms, such as Anderson Acceleration (Walker and Ni, 2011), that are adaptive to the problem constants. They can be seen as an automatic way to find the optimal combination of the previous iterates.
>
> ---
> Source: Azizian et al. (2020) , section 6.1.
>

In the same paper, the authors refer to Scieur et al. (2016) and Bollapragada (2018) for the theoretical analysis of such a method. In particular, the result presented in Bollapragada et al. (2018) proposes an analysis of Anderson Acceleration for *any* nonsymmetric operator $\textbf{G}$. They combine Theorem 3.1 (Crouzeix's bound) and Theorem 3.3. (optimal polynomial on ellipses from Fischer and Freund (1991), Th. 2), to obtain a general convergence bound. This result covers all the theoretical work presented in this paper.

Moreover, it is surprising that the author did not use the Crouzeix's bound for their Theorem 4.2. In short, the Crouseix's bound says that when dealing with a nonsymmetric matrix $\textbf{G}$ and polynomials $P_2$, we have

$$
|| P(G) ||_2 \leq c \max\_{\lambda\in W(\textbf{G})}  ||P(\lambda) ||_2,
$$
where $W(\textbf{G})$ is the *numerical range* of the matrix $\textbf{G}$, and $c$ is a constant, bounded by $2$ (Crouzeix's conjecture) and proven to be upper bounded by $1+\sqrt{2}$. In particular, it is unclear how the authors manage to prove the inequality (last display equation in the proof of Theorem B.2) (I removed the other details to focus on the important part)
$$
\min \_{f_p\in\mathcal{P}_p} ||f_p(\textbf{G}) || \leq \left(\frac{r}{p}\right)^p
$$
without using the Crouseix's bound, as $\textbf{G}$ is not symmetric.


That said, the numerical experiments are persuasive, but the paper should be presented in a whole other way to include the missing related work. I strongly recommend the author rework the paper and submit it to another venue, as Anderson mixing could positively impact the optimization of min-max problems.

**Summary Of The Paper:**

The authors propose to use the Anderson Acceleration on min max problem. They show two theoretical results: convergence rate on bilinear problems when using simultaneous gradient decent-ascent, and convergence rate on bilinear problems when using alternating gradient decent-ascent. Finally, they present numerical results in favor of their approach.

**Summary Of The Review:**

Because the original idea of the paper was briefly introduced in Azizian et al. (2020), but more importantly, because the theoretical results presented in this paper are already covered in Bollapragada (2018), I am not convinced the paper is novel enough to be published. Moreover, I have a concern about a detail in the proof of Theorem 2.

---

> ### Author Response · Authors · 2021-11-17
> **Response to reviewer 4 (part 1)**
>
> Thanks for your comments! In this response, we'll address your whole comment and explain the contributions of our work, which we believe has been misunderstood.
>
> ## 1: Mentioned by Azizian et al. (2020)
>
>  We kindly disagree with the remark that our work has limited novelty because it was first mentioned in Azizian et al. (2020). It is true that Azizian et al. (2020) shortly mentioned Anderson Mixing in Sec 6.1. However, the whole paper (Azizian et al. (2020)) mainly focuses on the convergence of momentum and has **NO** detailed formulation of GDA with Anderson Mixing, not to mention the lack of any theoretical results, numerical results, and codes regarding this addition. As Anderson Mixing could bring a positive impact on the community, this work fulfills the gap by investigating GDA with Anderson Mixing. We regard our work as the first and hopefully many excellent works to come. We admit this may be controversial and have already toned down our claim in the introduction section of the current revision.
>
> ## 2 The results in our paper has been covered by the results in [1]
>
> First, we would like to clarify that our theoretical results are **very different** from those in [1]. In particular, we found that the main results in [1] **may not be applicable**. They are based on applying the Fischer and Freund's theorem [2] to solve $\min _{p(z) \in \mathbb{C}[x]} \max _{p(1)=1}|p(z)|$ and obtain a convergent result.
>
>  Theorem 1 [Fischer and Freund's theorem [2]] says:
>
> Let $n \ge 5$ be an integer, $r>1$, and $c \in \mathbb{R}$. A constrained polynomial minmax problem can be solved uniquely by
> \begin{equation}
> t_{n}(z ; c):=\frac{T_{n}(z)}{T_{n}(c)},
> \end{equation}
> where
> \begin{equation}
> T_{n}(z)=\frac{1}{2}\left(v^{n}+\frac{1}{v^{n}}\right), \quad z=\frac{1}{2}\left(v+\frac{1}{v}\right)
> \end{equation}
> if
> (a) $|c| \ge \frac{1}{2}\left(r^{\sqrt{2}}+r^{-\sqrt{2}}\right)$ or
>
> (b) $|c| \ge \left(1 / 2 a_{r}\right)\left(2 a_{r}^{2}-1+\sqrt{2 a_{r}^{4}-a_{r}^{2}+1}\right),$
> where $a_{r}:=\frac{1}{2}\left(r+\frac{1}{r}\right)$.
>
> However, the polynomials considered in the analysis of [2] take value $1$ at the location $c=1$. This $c=1$ is included in $\mathscr{E}_{r}$ which **violates** the original assumptions of the Fischer and Freund's theorem. As a result, the upper bound obtained in [2] can be greater or equal to 1, which **can not guarantee** the convergence.
>
> Moreover, the results of [1] and [3] are very different from ours.  [1,3] analyze the convergence of Minimum polynomial extrapolation methods while our analysis focuses on Anderson Mixing. Anderson mixing is known as a quasi-Newton method [4] and the methods in [1,3] are both built on the Minimum polynomial extrapolation. Although the settings in these methods look very similar, there are indeed **some substantial differences**, which is also pointed out by [5,6, 7]. For example, [7] explicitly says
>
> *"Anderson acceleration differs mathematically from the vector-extrapolation methods and $\epsilon$-algorithms and, in fact, belongs to a distinct category of methods developed
> by a different community of researchers."*
>
> To be specific in our case, the iterates generated from Anderson Mixing are used directly in the fixed point iteration while two decoupled sequences of iterates are kept in Minimum polynomial extrapolation methods. This subtle difference will make the theoretical framework different.
>
> In addition, the minimax polynomial problem *considered in [1]* is
>
> \begin{equation}
> \min _{p \in \mathbb{C}[z] \atop p(1)=1} \max _{z \in W(A)}|p(z)|
> \end{equation}
>
> where $p(z)$ is a degree k polynomial with value 1 at 1 and the domain for maximum problem is the numerical range of iteration matrix $\mathbf{A}$.
>
> The minimax polynomial problem analyzed in *our proposed method* is
>
> \begin{equation}
> \min_{f_{p} \in P_{p}} \max_{x\in Sp(\mathbf{I}-\mathbf{G})} |f_{p}(z)|
> \end{equation}
>
> where $f_p$ is the residual polynomial for system $(\mathbf{I}-\mathbf{G})x = \mathbf{b} $ and the domain of polynomial is **not** numerical range but spectrum  of matrix $(\mathbf{I}-\mathbf{G})$.
>
> Furthermore, our nonasymptotic bound for the bilinear problem can compete with the asymptotic bound of EG with momentum, which is **better** than the bound in [1].
>
> Reference:
>
> [1] Raghu Bollapragada et al. “Nonlinear acceleration of momentum and primal-dual algorithms”
>
> [2] Bernd Fischer and Roland Freund. “Chebyshev polynomials are not always optimal”.
>
> [3] Damien Scieur et al. “Regularized nonlinear acceleration”
>
> [4] Haw-ren Fang and Yousef Saad. “Two classes of multisecant methods for nonlinear acceleration”
>
> [5] Claude Brezinski et al. “Shanks sequence transformations and Anderson acceleration”
>
> [6] Wei Fuchao et al. “Stochastic Anderson Mixing for Nonconvex Stochastic Optimization.”
>
> [7]  Homer F Walker and Peng Ni “Anderson Acceleration for fixed point iterations”

---

> ### Author Response · Authors · 2021-11-17
> **Response to reviewer 4 (part 2)**
>
> ## 3: Regarding why don't we use the Crouzeix's bound as the authors did in [1]?
>
> We also want to point out that a naive application of Crouzeix's bound to the minimax optimization problem **can not** be used to derive the convergent result. This is because the point $0$ where all the residual polynomials take the fixed value of $1$ is included in the numerical range of the iteration matrix, which violates the assumption of Fischer and Freund's theorem [2]. As a result, it can not be used to prove that the residual norm is decreasing based on this approach. Instead, we show that although the coefficient matrix is non-normal, it is diagonalizable. We then give the convergence results based on the eigenvalues instead of the numerical range.
>
> In Appendix B.1 in our revision, we give **detailed proof** and show that analysis based on numerical range **can not** help derive the convergent result as the upper bound is not guaranteed to be less than 1.
>
> ## 4: How did we prove the inequality in Theorem B.2 (B.3 in our updated version).
>
> In short, we use the eigendecomposition of $G$ and the special polynomial  $(\frac{c-t}{c})^{p}$ to derive the inequality in Theorem B.2.
> Since $G$ is diagonalizable (which has been shown in appendix), we assume the eigendecomposition of $G$ is $G = \mathbf{X}\Lambda\mathbf{X}^{-1}$. Then the last inequality comes from Lemma 6.26 and Proposition 6.32 in [8]. We have updated the proof to make it more clear in the Appendix in our revision.
>
> We hope our detailed response better explains the contributions of our work and clarify the difference between our work and [1]. We are eager to know if you have any concerns remaining; if your concerns have been clarified, we sincerely hope it helps you re-evaluate our paper.
>
>
> Reference:
>
> [8] Yousef Saad, Iterative methods for sparse linear systems

---

> > ### Comment · Reviewer_KCYs · 2021-11-18
> > **RE - Response to reviewer 4**
> >
> > I thank the author for their detailed response. I acknowledge I have underestimated the technical difficulty to derive the bounds of Theorem B2 (now B3). I will upgrade my rating after the discussion phase if the authors include in the revision the discussion about the numerical range.
> >
> > I will quickly comment on point 2): to my knowledge, the results in Bolapragrada et al. and Scieur et al. applies to Anderson Acceleration, not MPE (after a careful look at the algorithms in those papers). I believe the result of Bolapragrada et al. can be also used in the main paper to extend the theory to other settings that are not bilinear, for instance, bilinear-quadratic games such as $\min_x\max_y x^TAx - y^TBy + x^TCt + d$. This can strengthen a bit the discussion you made in Appendix B4 in the revision.

---

> > > ### Author Response · Authors · 2021-11-20
> > > **Thanks for your quick response. We have updated our revision.**
> > >
> > > We thank you for the quick response. We are delighted to hear you are willing to re-evaluate our work. We have incorporated your suggestion of adding the discussion about the numerical range in our revision (Section 4 page 5, B.1, and B.4). Appendix B.1 includes proof of the difficulty of deriving convergent results based on the numerical range for bilinear case. Appendix B.4 includes a discussion on deriving convergent results for quadratic case based on the numerical range.
> > >
> > > Here we quickly highlight what we added to the section:
> > > Consider a quadratic problem $\min_x\max_y x^TAx - y^TBy + x^TCy + d$. In short, we show that when both $A$ and $B$ are positive definite, analysis based on the numerical range leads to a convergence guarantee. However, when $A$ or $B$ is not positive definite, such a guarantee does not hold because 1 can be included and violates the assumption. In addition, we provided a convergence result for quadratic problems based on spectrum analysis. We have added this discussion in Appendix B.4 (page 22 & 23).
> > >
> > > First, we would like to re-emphasize the difficulty of analysis based on the numerical range [6], which we thank you for the reference. It studied an interesting topic where the underlying operator is non-symmetric and analyzes the convergence based on the numerical range. However, we found extending the existing results of Anderson Acceleration for (non) convex optimization to minimax optimization is *not trivial*. For example, it is easy to show that 1 is included in the numerical range for $G$ (the GDA-AM operator) for bilinear problems. On the other hand, 1 can be included in the numerical range for $G$ for quadratic problems when $A$ or $B$ is not positive definite, and thus there is no convergence guarantee, which is the reason why we didn't use numerical range as our analysis tool. Another example occurs in [1], which considers solving general nonsmooth fixed-point problems. They proposed a variant of Anderson Mixing that gives the first globally convergent algorithm assuming only that the fixed-point iteration is nonexpansive. However, the sim-GDA dynamic is a non-contractive mapping and the alt-GDA dynamic can only stay bounded [2]. As a result, applying results in [1] can not give us convergence results either. We believe a *key* contribution of our work is to show why Anderson Mixing helps transform a non-convergent sequence to a convergent one and derive its nonasymptotic rate from a spectral analysis perspective.
> > >
> > > Although this may be minor, we would like to reach an agreement on the difference between our work and [6] in this discussion phase. Vector extrapolation methods generate a new and *independent* accelerated sequence given the original sequence (fixed-point iterates). Different from vector extrapolation methods, Anderson Mixing uses the extrapolated point as the input of the fixed-point operator, where the word 'mixing' comes from. This subtle difference makes the memory table of them totally different.  Based on your recent comment, we assume we have reached an agreement that MPE is not equivalent to Anderson Mixing. After a close look at [6] and [3], we believe [6] can apply to Anderson Mixing but not the original RNA [3]. Although the updated arxiv version [4] changed it to Anderson Mixing (equation 5,6,7), the original RNA (neurips version[3]) explicitly says its base extrapolation algorithm is minimal polynomial extrapolation (MPE).  The wording in [6] is confusing as it states `it leads to the (RNA) [3]' and the inputs to algorithm 1 are the original iterates generated by fixed-point (which is different from [1,5]).  However, after checking the detailed proof, we found the analysis in [6] actually applies to Anderson Acceleration.  We believe analysis based on the numerical range [6] is inspiring and can be helpful to investigate the convergence results of quadratic problems as you suggested. We have added a discussion on the numerical range and explained its applicability of applying it to minimax optimization in Appendix B.1 (bilinear) and B.4.1 (quadratic).
> > > Another difference of contribution we would like to stress here is that the analysis in [6] assumes a full memory version and ours assumes a restarted version, which is more common in practice.
> > >
> > > Thank you for suggesting the discussion on the quadratic games. Thank you for your willingness to raise your rating. Please let us know if you have any questions or concerns so we can further strengthen our paper.
> > >
> > >
> > > [1] Junzi Zhang et, al. `Globally Convergent Type-I Anderson Acceleration for Non-Smooth Fixed-Point Iterations'
> > >
> > > [2] Guodong Zhang et, al. `Near-optimal Local Convergence of Alternating Gradient Descent-Ascent for Minimax Optimization'.
> > >
> > > [3] NeurIPS RNA
> > >
> > > [4] Updated arxiv version RNA `https://arxiv.org/abs/1805.09639'
> > >
> > > [5] Brezinski et, al. `Shanks Sequence Transformations and Anderson Acceleration
> > >
> > > [6] Bollapragada et, al. `Nonlinear Acceleration of Primal-Dual Algorithms'

---

> ### Author Response · Authors · 2021-11-28
> **Look forwarding to hearing back from you before the discussion phase ends**
>
> Dear reviewer,
>
> Thanks again for your time and detailed review. We are wondering if we misunderstand your meaning since we believe it is impossible to update the score after the discussion phase, which is the meta review phase. We are encouraged that after our first response, you mentioned you will upgrade the rating after the discussion phase if the authors include in the revision the discussion about the numerical range.
>
> Our work has been strengthened a lot by adding additional discussions on the numerical range and incorporating suggestions from other reviewers. Analysis based on the numerical range is interesting and we will continue study it for minimax problems in future work. We appreciate for your precious time reviewing this work.
>
> Best,
>
> Paper2078 Authors

---

> ### Author Response · Authors · 2021-11-29
> **Thank you for your support**
>
> Thank you for re-evaluating our work and updating the score, although you may have forgotten about the technical score. We appreciate the discussion with you.

---

### Official Review · Reviewer_VjNk · 2021-11-01

**Correctness:** 3
**Technical Novelty And Significance:** 3
**Empirical Novelty And Significance:** 3
**Recommendation:** 6
**Confidence:** 4

**Main Review:**

## Strengths:
- The method is well motivated and the paper is well written.
- The theoretical results are sound
- The method seems to scale to GANs (when combined with Adam)

## Weaknesses:
- The comparison with the related work could be improved (see my detailed comment)



I think the authors could discuss more the rates obtained in Thm 4.2 and Thm 4.1 and the standard rates obtained in the literature (e.g. the ones presented in Remark 4.2). For instance, a first step would be to stress that the value of $T_p(1 + x)$ has a closed-form solution (below Equation 21). Also, one could compute the Taylor expansion of $T_p(1 + x)$ when $x\to 0$ to have an idea of the convergence speed. Looking at the formula below (21) It seems that you get an accelerated convergence rate (i.e. $\rho(A)^{1/p} = 1 - O(1/\sqrt{\kappa})$. However, I am a bit surprised that you get this rate for $p=1$ because in that case, GDA-AM corresponds to Gradient Descent with line-search. Can you comment on the formula for the expression of the Chebyshev polynomial of degree p at the point $1+ \frac{2}{\kappa-1}$?

For the experiments (Fig 4): Azizian et al. 2020 use *positive* momentum + EG. I would be interested to see if SIM-AM still outperforms EG + *positive* momentum. Also, how did you tune the step size and the other hyper-parameters of the methods for the results obtained in figure 5?
Moreover it seems counter intuitive since EG + positive momentum has a convergence rate of $\rho(A)^{1/p} = 1 - O(1/\sqrt{\kappa})$

## Summary of the Questions asked (by decreasing order of importance):
1. Can you comment on the formula for the expression of the Chebyshev polynomial of degree p at the point $1+ \frac{2}{\kappa-1}$? It seems that your formula gives $\rho(A)^{1/p} = 1 - O(1/\sqrt{\kappa})$ while for $p=1$ it should be $\rho(A)^{1/p} = 1 - O(1/\kappa)$
2. Can you compare your method with EG + positive momentum? (Empirically)
3. Theorem 4.2 seems to give a convergence rate of $1 - O(1\kappa)$  Do you agree with this statement?

## Minor Question
From my understanding, in order to get the theoretical guarantees you do not even need to use Anderson Mixing at each time step but only every p timesteps (and you only need to ensure that linear combination of $x_{t-i}$ can generate any polynomial of degree $p$ with coefficients that sums to 1).
Is it why you analyze the algorithm every p step?
What is the computational overhead (solving each timestep vs solving each p steps?)
Do you consider this algorithm because it is more likely that it will extend beyond the least-square case?


**Summary Of The Paper:**

This paper proposes a new optimization method for min-max optimization (That could actually be generalized to any variational inequality problem) based on Anderson Mixing. They prove the convergence of their algorithm in the case of bilinear min-max games for both alternated and simultaneous updates operator. They eventually try their algorithm on toy bilnear problems and GANs.

**Summary Of The Review:**

As a summary, I think this submission is well written. I think some more work could be done to compare the obtained rates with the ones already present in the literature but it should be relatively easy to implement.  Also, I have some concerns regarding the theory and the comparison with respect to the baselines.

I will increase or decrease my score depending on how the authors address my comments. (for instance, if everything is clarified I will increase the technical and empirical novelty score as well as the final score)

For now, I recommend accepting this paper.

---

> ### Author Response · Authors · 2021-11-17
> **Response to reviewer 3 (part 1)**
>
> We would like to thank the reviewer for the time and detailed review. We are encouraged by your positive remarks on our theoretical developments and empirical results. We found your questions great and important, so we have added a discussion section in the Appendix. Please see our answers to your questions below (in two parts).
>
>
> ### 1: Explanation on why taking Chebyshev polynomial of degree p at the point $1+\frac{2}{\kappa-1}$
>
> We evaluate the Chebyshev polynomial at this specific point because the reciprocal of this value gives the **minimal** value of infinite
> norm of the all polynomials of degree $p$ defined on the interval based on Theorem 6.25 on [1]. In other words, taking the function value at this point leads to a **tight** bound.
>
> ### 2: Discussion on comparison with existing bounds
>
> First of all, GDA-AM doesn't correspond to Gradient Descent with line-search when $p=1$. When $t\geq 1$, $T_p(t)= \frac{1}{2}((t+\sqrt{t^2-1})^p+(t+\sqrt{t^2-1})^{-p})$. Thus, when $p=1$, $T_1(t)=t.$ In this case, our bound $\rho(A)=(\frac{1}{1+\frac{2}{\kappa(A^TA)-1}})^2 = (\frac{\kappa(A^TA)-1}{\kappa(A^TA)+1})^2$. This shows that when $p=1$, $\rho(A)= (1-\frac{2}{\kappa(A^TA)})^2$ or $1-O(\frac{1}{\kappa{(A^TA)}})$. This bound takes a different form compared with existing bounds because of the appearance of the exponent $2$.
>
> Alternatively, we can also derive another bound for comparison with existing bounds. If we use the inequality that $T_p(t)\geq \frac{1}{2}((t+\sqrt{t^2-1})^p)$, we can obtain the bound $\rho(A)=4(\frac{\sqrt{\kappa(A^TA)}-1}{\sqrt{\kappa(A^TA)}+1})^2=4(1-O(\frac{1}{\sqrt{\kappa(A^TA)}}))$, which is in a form that is comparable with EG and can compete with EG + positive momentum. The numerical experiments in figure 2b numerically verify that our bound is smaller than EG. We wanted to compare our rate with EG with positive momentum. However the bound of EG with positive momentum is asymptotic and does not specify the constants $c$, we can not numerically compare them. We do provide an empirical comparison between GDA-AM and EG with positive momentum for bilinear problems in Appendix C.1. It shows GDA-AM outperforms EG with positive momentum. A detailed discussion is now included in Appendix B.3 in the revision.
>
> ### 3: If SIM-AM still outperforms EG + positive momentum?
>
> Yes. We use EG with negative momentum because we found it is easier to tune and tends to have better performance. We have added the numerical comparison with EG + positive momentum in the revision. It still indicates our method converges faster in terms of iteration number. Since the accelerated convergence of EG with positive momentum is **asymptotic** and requires all three hyperparameters to satisfy certain conditions. We used the best step size and momentum we could find through grid search because it was really hard for us to find a good combination. In contrast, it shows the advantages of GDA-AM. Our accelerated convergence is **nonasymptotic**. Further, it **does not** require a **strict condition** for step size; any reasonable step size that does not cause overflow will converge fast. Please let us know if you have any recommendations on choices of hyperparameters of EG with positive momentum. And we will update experiments according to your suggestion.
>
> ### 4: How did you tune the step size and the other hyper-parameters of the methods for the results obtained in figure 5?
>
>  Figure 5 depicts the convergence in terms of time for the same run used in Figure 4 for fairness. That's saying the hyperparamters, problems, initializations and results of figure 4 and 5 are all the same with the only difference of x-axis. We set the step size as 1 for all methods except simGDA and OG because simGDA diverges. The step size of OG is 0.5 because theoretically it should be half of of the step size of EG and OG tends to diverge for $\eta>0.5$. The momentum term for EG+NM is set as -0.3, which turns out to be the most robust and aggressive one we found. The table size $p$ is set as 10. Indeed GDA-AM converges fast for any $p>1$, we use 10 because it requires small computation overhead.
>
> ### 5: Theorem 4.2 seems to give a convergence rate of $1-O(\frac{1}{\kappa(A^TA)})$? Do you agree with this statement?
>
>  We do not agree with this statement. This is because the bound in Theorem 4.2 depends on the eigenvalue distribution of the matrix $\mathbf{G}$. Condition number is not directly related to the distribution of eigenvalues of a nonsymmetric matrix $\mathbf{G}$. Thus, the condition number is not a precise metric to characterize the convergence. If these eigenvalues are clustered, then our bound can be small. On the other hand, if these eigenvalues are evenly distributed in the complex plane, then the bound can be very close to $1$.

---

> > ### Comment · Reviewer_VjNk · 2021-11-29
> > **Thank you for your answer**
> >
> > thank you for your answer.
> > I apologize for getting back to the discussion a bit late.
> >
> > 2. I think it would improve the presentation of the method if the author could include why this method does not correspond to GD with line search after the first iteration.
> >
> > 2. I strongly disagree with the statement that $\rho(A) = 4(1- O(1/\sqrt{\kappa}))$ is comparable with EG and EG+ (or even GD). What we care about is, how close is $\rho$ from 1. With this bound $\rho(A) = 4(1- O(1/\sqrt{\kappa}))$ , $\rho$ is actually close to 4 which implies divergence.
> >
> > 5. The statement on the bound of the eigenvalues of $G$ is a bit weak. I feel that because of its particular bloc structure of the matrix $G$, the eigenvalues $G$ can most likely be bounded by a function of $\kappa$
> >
> > Note that the caption Fig 3 is currently overlapping with Theorem 4.2.
> >
> >
> > Overall I think this paper is borderline in terms of presentation and contribution. However, since the consensus of the other reviews is mainly positive. I will maintain my score

---

> > > ### Author Response · Authors · 2021-11-30
> > > **Thanks for your reply. We hope this reply can better clarify any misunderstanding.**
> > >
> > > Dear reviewer,
> > >
> > > We still thank you for the response. We hope this reply can better clarify any misunderstanding.
> > >
> > > **2:**  We would like to highlight that Anderson mixing and line search are very different. In the literature of Anderson Mixing, it's rare to see $p=1$ while $p\geq2$ is a common setting. Instead, [1] show that Anderson Mixing can be viewed as a multi-secant quasi-newton method when $p\geq2$. Thanks for your suggestion, we will definitely discuss why GDA-AM is very different from GDA with line search.
> > >
> > > **3:** We provide a **tight** and **convergent** bound in our main paper, **not**  $\rho(A)=4(1-O(\frac{1}{\sqrt{\kappa(A^TA)}}))$. We acknowledge $\rho(A)=4(1-O(\frac{1}{\sqrt{\kappa(A^TA)}}))$ is not comparable with existing rates and that is the reason why we **didn't use this bound** in our paper. **We provide this form only in our rebuttal to address your concern(comparing with other rates).**  In our previous response, we wanted to show that the convergent bound in our main paper takes a different form compared with existing bounds and there exists another form that has a similar form with existing bounds (not comparable, sorry for using a wrong word). As a result, **we numerically verify that our bound is smaller than EG in figure 2b.** We wanted to compare our rate with EG with positive momentum. However the bound of **EG with positive momentum is asymptotic, requires strong assumptions and does not specify the constants**, we can not numerically compare them. We do provide an empirical comparison between GDA-AM and EG with positive momentum for bilinear problems in Appendix C.1. It shows **GDA-AM outperforms EG with positive momentum**.
> > >
> > >
> > > **4:** We agree with the statement that eigenvalues $G$ can most likely be bounded by a function of $\kappa$. But we believe the bound of Alternating GDA-AM based on $\kappa$ **can not** be characterized
> > >  **in a unified and concise form**. In the proof of Theorem 4.2 (Appendix B.2), we show that the eigenvalues of $G$ are related to the singular values of $A$ due to the block structure in the following way:
> > >
> > > $\lambda_{\pm i}=\frac{\left(\eta \sigma_{i}\left(\eta \sigma_{i} \pm \sqrt{\left(\eta \sigma_{i}\right)^{2}-4}\right)\right)}{2}$
> > >
> > > where $\sigma_{1}\ge \sigma_{2}\ge\dots\ge \sigma_{n}$ are the singular values of $A$. It's easy to see that $\mathcal{R}\left(\lambda_{\pm i}\right) \geq \frac{\left(\eta \sigma_{i}\right)^{2}}{2}$. When $\eta\sigma_{i} \le 2$, the imaginary part of eigenvalues is not zero.
> > >
> > > This formula indicates that G can have real eigenvalues as well as complex eigenvalues depending on the learning rate and the magnitude of the singular values. For example, when the learning rate is extremely large such that $\eta \sigma_i$>2 for all $\sigma_i$, we can derive a bound related to the condition number of A, but **this is not realistic**. In more realistic setting the analysis is very complicated due to the appearance of imaginary part of the eigenvalues of G. A discussion of this can be easily added in our updated version.
> > >
> > > Please let us know if you still have any concerns.
> > >
> > > Best,
> > >
> > > Paper2078 Authors
> > >
> > >
> > > [1] Two Classes of Multi-secant Methods for Nonlinear Acceleration

---

> > > > ### Comment · Reviewer_VjNk · 2021-11-30
> > > > **Thanks for your quick answer**
> > > >
> > > > 3: Figure 2b is pretty impressive. What what the dimensionality of the matrix ?
> > > >
> > > > 4: if $\eta \sigma_i \leq 2\,,\, \forall i$ then we have $\lambda_i = \frac{\eta^2 \sigma_i^2 \pm i \eta \sigma_i\sqrt{4-\eta\sigma_i}}{2}$. Then, it should not be too hard ot show that the limiting pair of eigenvalues is $\frac{\eta^2 \sigma_\min^2 \pm i \eta \sigma_\min\sqrt{4-\eta\sigma_\min}}{2}$
> > > >
> > > > Then you could solve
> > > > $$\frac{\eta^2 \sigma_\min^2 \pm i \eta \sigma_\min\sqrt{4-\eta\sigma_\min}}{2} \in B(c,r)
> > > > \Leftrightarrow |\frac{\eta^2 \sigma_\min^2 \pm i \eta \sigma_\min\sqrt{4-\eta\sigma_\min}}{2} -c|^2 \leq r^2$$
> > > >
> > > > Minimizing the equality case with respect to $r/c$ (with the constraint that $c\leq r \leq 0$) should get you a closed-form solution for $r/c$

---

> > > > > ### Author Response · Authors · 2021-11-30
> > > > > **Re: Thanks for your quick answer. Response to the bound in Theorem 4.2**
> > > > >
> > > > > Thanks for your feedback. We are happy to further discuss with you on this.
> > > > >
> > > > > **3:** We use 100 in figure 2b. We can observe similar patterns when the dimension is larger than 100.
> > > > >
> > > > > **4:** There is a small typo in the formula of the eigenvalues in your response. It should be $\lambda_{i}=\frac{\eta^{2} \sigma_{i}^{2} \pm i \eta \sigma_{i} \sqrt{4-\eta^2 \sigma^2_{i}}}{2}$ when  $\eta\sigma_i <2$ for all $i$. We greatly thank you for your suggestion! We agree with the statement that eigenvalues  can most likely be bounded by a function of $\kappa$ under certain conditions. But in fact, we need to consider three different cases:  1) $\eta\sigma_i <2$ for all $i$. In this case, all the eigenvalues of $G$ have non-zero imaginary parts. 2) $\eta\sigma_i >2$ for all $i$. In this case, all eigenvalues of $G$ are all real. 3) $\eta\sigma_i < 2$ for some eigenvalues. The eigenvalues of $G$ can have both real and complex eigenvalues.
> > > > >
> > > > > In all three cases, the circle $B(c,r)$ should enclose all eigenvalues, in particular the leftmost ones corresponding to the smallest singular value $\sigma_{min}$, the rightmost eigenvalues corresponding to the largest singular value $\sigma_{max}$ and **the eigenvalues with the largest imaginary part**.  In the response, it seems that you only consider the leftmost eigenvalues. We kindly argue that just minimize the equality your mentioned above can not guarantee the circle would also enclose the rightmost eigenvalues of G and the ones with largest imaginary part. Also the circle should exclude the origin. Although it is very tempting to try to express things in terms of spectral condition numbers ($\sigma_{max}/\sigma_{min}$), we honestly don't believe it is that simple. Different from the theoretical result of Sim GDA-AM (Theorem 4.1), **we aim to provide a general bound of alternating GDA-AM in a unified and concise form** (Theorem 4.2), with a similar bound also considered in Corollary 6.33 in [1].
> > > > >
> > > > > [1] Iterative Methods for Sparse Linear Systems Second Edition
> > > > >
> > > > > We also provide the Matlab code to plot the eigenvalues of G in different settings (case 3 here). It can be observed that the eigenvalues with the largest imaginary part usually do not correspond to the smallest eigenvalues. This is because the value under the square root is multiplied by $\eta\sigma_i$, that is the imaginary part of $\eta\sigma_{min}\sqrt{4-(\eta\sigma_{min})^2}$ might be smaller than that of $\eta\sigma_i\sqrt{4-(\eta\sigma_i)^2}$ for some $i$, which makes the analysis complicated.
> > > > >
> > > > >     n = 3000;
> > > > >     lamda = zeros(2*n,1);
> > > > >     %% you can choose different value to simulate the range of $\eta*\sigma_i$%%
> > > > >     x = linspace(0, 4,n);
> > > > >     %%This is case 3, The eigenvalues of  G can have both real and complex eigenvalues.%%
> > > > >     %%x = linspace(0, 1.9, n) for case 1, all the eigenvalues of G have non-zero imaginary parts.%%
> > > > >     %%x = linspace(2.1,4, n) for case 2, all eigenvalues of G are all real%%
> > > > >     for i = 1:n
> > > > >         lamda(i) = x(i)^2 + x(i)*sqrt(x(i)^2 - 4);
> > > > >         lamda(i+ n) =  x(i)^2 - x(i)*sqrt(x(i)^2 -4);
> > > > >     end
> > > > >     scatter(real(lamda),imag(lamda))

---

> ### Author Response · Authors · 2021-11-17
> **Response to reviewer 3 (part 2)**
>
> Following part 1
>
> ### 6: From my understanding, in order to get the theoretical guarantees you do not even need to use Anderson Mixing at each time step but only every p time steps (and you only need to ensure that linear combination of can generate any polynomial of degree with coefficients that sums to 1). Is it why you analyze the algorithm every p step?
>
> In order to guarantee the global convergence of GDA-AM, we need to apply Anderson Mixing every time step. In order to control the computational overhead, we restart Anderson mixing every p steps. Let's say there are three implementations of Anderson Mixing, 1: increasing table size. 2: fixed table size $p$. 3: restart version (our choice). Obviously, the first choice, Anderson mixing with increasing table size is unfavorable and can even cause ill conditions. The second choice, Anderson mixing using a fixed table size $p$ updates the table by deleting the oldest iterate and adding the new coming iterate. This is still not optimal. When the time step $t>p$, this implementation has a fixed memory and computational overhead during the whole training procedure. And restarted Anderson Mixing cleans up the table every $p$ iterations and thus achieves lower computational overhead.
>
> ### 7:  What is the computational overhead (solving each time step vs solving each p steps)?
>
> The computational overhead of restarted Anderson comes from solving a $p_t\times p_t$ linear system to get the mixing coefficient $\beta_i$ at time step because the table size $p$ is changing every iteration. The cost of updating the coefficient matrix and solve the linear system is $O(p_t^2n)$, which scales linearly at each time step. Please let us know if we misunderstood your question.
>
> ### 8: Do you consider this algorithm because it is more likely that it will extend beyond the least-square case?
>
> This is because Anderson acceleration method can be extended to the nonlinear problems. As also suggested by [2], Anderson Mixing can be applied to sequences not belonging to the Shanks kernel, which indicates Anderson Mixing is a more general framework.
>
> Again, we thank the reviewer for appreciating our contributions and asking great questions. These very detailed questions let us realize it is necessary to discuss more our obtained rates. We have incorporated answers to your questions as well as our technical contributions in the discussion section in Appendix B.3. We'd like to get in touch with you again to know if we clarified your concerns and questions.
>
> Reference:
>
> [1] Yousef Saad, Iterative Methods for sparse linear systems, Second Edition.
>
> [2] Claude Brezinski, Shanks and Anderson-type acceleration techniques for systems of nonlinear equations, 2021

---

> ### Author Response · Authors · 2021-11-28
> **We are glad to discuss further comments and suggestions before the discussion phase ends**
>
> Dear reviewer,
>
> Thanks again for your time and helpful comments.  Since you mentioned you will increase or decrease the score depending on how we address your comments, we believe our response and updated draft can address your comments and concerns. Please let us know if you have further questions.
>
> Further, we strengthened our paper by incorporating suggestions from all reviewers, i.e., additional experimental comparisons and discussions. We feel this complements perfectly with the positive results already been carried out, and they together attribute to a better work. We are eager to hear back from you if you have any feedback or further questions.
>
> Best,
>
> Paper2078 Authors

---

### Official Review · Reviewer_t7gv · 2021-11-02

**Correctness:** 4
**Technical Novelty And Significance:** 3
**Empirical Novelty And Significance:** 2
**Recommendation:** 8
**Confidence:** 3

**Main Review:**

1) From the theoretical point of view, multiple analyses are missing. The only results are for GDA and bilinear cases. The authors didn't provide any analysis of their framework for the convex-concave case.

2) Authors didn't provide any result for the Nonconvex-Nonconcave case( convergence to $\epsilon$ stationary point). I think the result in [1] can be used to provide guarantees in this case. This is necessary to show the effectiveness of the method for GAN-related applications(e.g., the one in section 6.2).

3)Author motivates the subject from a theoretical and practical point of view.

4)In the introduction, the author reviewed the previous works carefully.

5)The authors successfully provide simulation results for their algorithm.

6)The paper is well-organized and well-written.

7)the code is included for the bilinear case, but I couldn't find the code for the image generation experiment.


[1]Wei, Fuchao, Chenglong Bao, and Yang Liu. "Stochastic Anderson Mixing for Nonconvex Stochastic Optimization." arXiv preprint arXiv:2110.01543 (2021).
-------------------------------------------------------------------
Post rebuttal :
I have read the authors response, and I increase
my score, but they should add the result on
general convex-concave problem and non-concave
setting and the discussion about it in the final version.
Also, I think there is still room to get faster convergence
 to saddle point that would be interesting if they can add
to the final version.
------------------------------------------------------------------

**Summary Of The Paper:**

This paper focuses on solving minimax optimization using Anderson mixing. Anderson mixing is a framework to accelerate fixpoint iteration. The authors tried to use  Anderson mixing for the minimax optimization problem. In their framework, they use a weighted average of the solution of last p iterates and optimized over the weights to come up with a new point.
They show that for the bilinear case, this algorithm converges to the optimal stationary point. Finally, the authors simulated their method on synthetic examples as well as image generation on CIFAR10.

**Summary Of The Review:**

1) From the theoretical point of view, multiple analyses are missing. The only results are for GDA and bilinear cases. The authors didn't provide any analysis of their framework for the convex-concave case.

2) Authors didn't provide any result for the Nonconvex-Nonconcave case( convergence to $\epsilon$ stationary point). I think the result in [1] can be used to provide guarantees in this case. This is necessary to show the effectiveness of the method for GAN-related applications(e.g., the one in section 6.2).

3)Author motivates the subject from a theoretical and practical point of view.

4)In the introduction, the author reviewed the previous works carefully.

5)The authors successfully provide simulation results for their algorithm.

6)The paper is well-organized and well-written.

7)the code is included for the bilinear case but I couldn't find the code for the image generation experiment.


[1]Wei, Fuchao, Chenglong Bao, and Yang Liu. "Stochastic Anderson Mixing for Nonconvex Stochastic Optimization." arXiv preprint arXiv:2110.01543 (2021).

---

> ### Author Response · Authors · 2021-11-16
> **Response to reviewer 2**
>
> We thank you for detailed feedback and suggestions. We really appreciate you found our work to be well-motivated and has empirical significance. Your comments and additional reference improved our paper. We have incorporated your suggestions and added theoretical results for both convex-concave and convex-nonconcave problems, which we think it addressed your main concerns.  In short, we show that GDA-AM converges to local Nash for general convex-concave problems. Given the time constraint, we spent most of our time on convergence analysis of stochastic convex-nonconcave problems. Thanks for the excellent reference, we are able to obtain a rate of
> $O(\epsilon^{-4})$. This result is satisfactory because we found the state of art rate of the deterministic convex-nonconcave problem is $O(\epsilon^{-3})$.  Although our new proof is based on [1], there is also a key difference with [1]. Here we are considering minimax optimization problem. Thus our gradient is actually $V(\mathbf{w})=\begin{bmatrix} -\nabla_{\mathbf{x}} f(\mathbf{x}, \mathbf{y}) \ \nabla_{\mathbf{y}} f(\mathbf{x}, \mathbf{y}) \end{bmatrix} $ rather than $
> \nabla f(\mathbf{w})  = \begin{bmatrix} \nabla_{\mathbf{x}} f(\mathbf{x}, \mathbf{y}) \ \nabla_{\mathbf{y}} f(\mathbf{x}, \mathbf{y})\end{bmatrix}$. This key difference indeed introduces some technical difficulties to show the convergence. Due to page limitations, we defer this result to the Appendix in the revision. And to our best knowledge, this is the first work to show the convergence of GDA with Anderson Mixing for stochastic convex-nonconcave problems.
>
> In addition, we have uploaded the codes for GAN's image generation experiments as well.
>
> Further, we would like to stress several technical contributions that might have been missed:
>
>
> **1**: Our obtained Theorem 4.1 and 4.2 provide **nonasymptotic** guarantees, while most other work are asymptotic. Recent work like EG with positive momentum can achieve an **asymptotic** rate of $1-O(1/\sqrt{\kappa})$ under strong assumptions (3 hyperparamteres).
>
> **2**: As we also noted to R1, our contribution is not just about to fix the convergence issue of GDA by applying Anderson Mixing; another contribution is that we arrive at a convergent and tight bound on the **original work** and not just adopting existing analyses. We developed Theorem 4.1 and Theorem 4.2 from a new perspective because applying existing theoretical results (i.e, GMRES and Crouseix's bound mentioned by R4) fail to give us neither convergent nor tight bounds.
>
> **3**: The novel Theorem 4.1 only requires *mild conditions* and reflects how the table size $p$ controls the convergence rate. It is also **independent** of the learning rate $\eta$. However, the convergence results of other methods like EG and OG **depend** on the learning rate, which may yield less than desirable results for ill-specified learning rates.
>
> We are eager to hear back from you if you have any feedback or further questions.
>
>
> Reference:
>
> [1] Fuchao Wei, Chenglong Bao, and Yang Liu. “Stochastic Anderson Mixing for Nonconvex Stochastic Optimization”

---

> ### Author Response · Authors · 2021-11-20
> **Further response: thank you for the quick response and increasing the score.**
>
> Thanks for reading our response and being willing to raise the score. Your suggestion and reference definitely strengthen our work. Since we have provided several experiments for general settings in the appendix, we will work on more challenging problems and provide numerical results in the future.  Regarding faster convergence to saddle point, we believe it is already a tight bound for bilinear problems. So we will work on investigating further the potential faster rate for general cases. We promise that we will update the paper according to your suggestions in the camera-ready version should our paper be accepted.

---

### Official Review · Reviewer_gBHU · 2021-11-03

**Correctness:** 3
**Technical Novelty And Significance:** 2
**Empirical Novelty And Significance:** 2
**Recommendation:** 5
**Confidence:** 3

**Main Review:**

Strengths:
S1 The authors propose using Anderson mixing strategy to solve the minimax optimization problem.
S2. They theoretically show that the proposed method can achieve global convergence guarantees under mild conditions.
S3. They conduct numerical experiments on some minimax problems for the CIFAR10 and Celeb A datasets to show that the proposed method converges faster for some convex-concave and non-convex-concave functions.


Weaknesses:
W1. The authors only discuss the convergence results for the simple bilinear games problems. They do not discuss the general nonconvex-nonconcave problem or the convex-nonconcave problem.

W2. The Anderson acceleration technique is a standard method for accelerating the fix-point procedures. The algorithms developed in this paper seem incremental.

W3. Many important references in this field are missing (See below). The authors should make numerical comparisons or discuss the differences between these methods.

References:
[1] A Single-Loop Smoothed Gradient Descent-Ascent Algorithm for Nonconvex-Concave Min-Max Problems. NeurIPS 2020.
[2] An accelerated inexact proximal point method for solving nonconvex-concave minmax problems. SIOPT 2021.
[3] On gradient descent ascent for nonconvex-concave minimax problems. ICML 2020.
[4] Solving a Class of Non-Convex Min-Max Games Using Iterative First Order Methods. NeurIPS 2019.
[5] Efficient Search of First-Order Nash Equilibria in Nonconvex-Concave Smooth Min-Max Problems. SIOPT 2021.


**Summary Of The Paper:**

Minimax optimization is a central problem in machine learning. The gradient descent ascent method is the most commonly used algorithm to solve this problem. This paper views the gradient descent ascent method as a fixed-point iteration and solves it using Anderson Mixing to converge to the local minimax. They show theoretically that the algorithm can achieve global convergence for bilinear problems under mild
conditions. Some numerical experiments have been conducted.


**Summary Of The Review:**

See above.

---

> ### Author Response · Authors · 2021-11-16
> **Response to reviewer 1**
>
> We greatly appreciate your remarks on our theoretical developments and empirical results. We understand your concerns about W1) a lack of general minimax problem analysis, W2) the algorithm itself is simple, and W3) missing necessary references. We have tried our best to improve our work by incorporating your suggestions (W1, W3) and clarifying our technical contribution (W2). We hope it helps you appreciate our result better:
>
> ### W1. The general nonconvex-nonconcave problem or the convex-nonconcave problem.
>
> We understand your concern and we believe adding analysis of general settings will definitely strengthen our work. We want to let you know that we've incorporated your suggestion. We obtained a convergence guarantee for convex-concave games and a convergence rate of $O(\epsilon^{-4})$ for stochastic convex-nonconcave problems. Given page limitation, it is not easy to add space for introducing the notation of convergence for general problems (i.e, $\epsilon$ stationary point) in the main paper.  We present this result in the appendix in our revision. We hope this analysis adds value to our work.
>
>
> ### W2. The algorithms developed in this paper seem incremental.
>
> We understand your point and believe this is not your main concern. But, please allow us to further clarify our technical contributions here. We believe applying Anderson Mixing into GDA is simple and direct yet powerful and its simplicity is one advantage of our algorithm. Further, it is a first and important step to explain the effectiveness of Anderson Mixing for minimax problems. Our framework is general and leaves scope for future work to use as a hammer on general settings with novel implementations. More importantly, we would like to emphasize that our contribution is not solely just about fixing the convergence issue of GDA by applying Anderson Mixing; our contribution is also that we arrive at a convergent and tight bound on the **original work** and not just adopting existing analyses. For example, we developed Theorem 4.1 and Theorem 4.2 from a new perspective because applying existing theoretical results (like Crouseix's bound mentioned by R4 and existing GMRES results) **can not give us convergent or tight bounds.**  Compared to GDA, extra-gradient, and extrapolation techniques on general optimization, Anderson Mixing on minimax optimization are harder to analyze. Indeed, convergence results of Anderson Mixing is not necessarily guaranteed to converge to local Nash and thus it is unclear whether GDA with Anderson Mixing is convergent or not. To overcome this difficulty, we introduced the machinery from the numerical analysis community to solve this problem. To be more specific, we study the spectrum of the GDA operator in a more detailed way so that we essentially showed Anderson mixing can improve the spectrum of the minimax game operator. We also added a stochastic version algorithm and corresponding theoretical analysis for general convex-(non)concave.
>
> In addition, it is worth highlighting another technical contribution; the **nonasymptotic** convergence rate of Algorithm 1 is **independent** of the learning rate $\eta$ while the **asymptotic** convergence results of other methods like EG and OG depend on the learning rate, which we think is another big step.
>
> ### W3. More related works.
>
> Thanks for your suggestions. We have already added those related works and a discussion on the differences between these methods in the rebuttal revision. Please see section 5 (related work) in our revision. Since the experimental setting in these related works are interesting but different from the focus of our work, we added additional comparison with EG + positive momentum (as suggested by R3) in Appendix C.1 in our revision . Our method still outperforms EG + positive momentum.
>
>
> We hope our detailed response and added theoretical results better explains the contribution of our work. If there are any remaining questions/concerns that make you hesitate to raise the score, we would be grateful if the reviewers raise them so that we could further improve our work.

---

> ### Author Response · Authors · 2021-11-23
> **Further response: add experimental comparison with mentioned references**
>
> We want to let you know that in addition to adding analysis for more general settings, we added a numerical comparison of GDA-AM with [1,2,3,4] by training a robust neural network on the MNIST data set against adversarial attack. We provide the results and experimental details in Appendix C.2 in our latest revision. The test accuracies of GDA-AM  under Fast Gradient Sign Method (FGSM) [5] and Projected Gradient Descent (PGD) attack [6] compared with other algorithms are given in the table below. We can observe that our results are comparable with the state-of-the-art methods and outperform them in several cases. In addition, GDA-AM doesn't exhibit a significant drop in accuracy when $\epsilon$ is larger and this suggests the learned model is more robust.
>
>  We believe we have addressed your concerns, especially regarding the general analysis and comparison with other methods. Indeed, the convergence properties of GDA-AM for general minimax problems are hard to analyze because existing theoretical results are inapplicable for minimax problems. We plan to continue investigating GDA-AM for more general settings in future works. Although we cannot further update our draft, please let us know if you have any questions and we will answer them and update the paper according to your suggestions.
>
> |         	| Natural 	|        	| FGSM   	|        	|        	| PGD    	|        	|
> |:-------:	|---------	|:------:	|--------	|--------	|--------	|--------	|--------	|
> | $\epsilon$|         	| 0.2    	| 0.3    	| 0.4    	| 0.2    	| 0.3    	| 0.4    	|
> |   [1]   	| 98.58%  	| 96.09% 	| 94.82% 	| 89.84% 	| 94.64% 	| 91.41% 	| 78.67% 	|
> |   [2]   	| 97.21%  	| 96.19% 	| 96.17% 	| 96.14% 	| 95.01% 	| 94.36% 	| 94.11% 	|
> |   [3]   	| 98.20%  	| 97.04% 	| 96.66% 	| 96.23% 	| 96.00% 	| 95.17% 	| 94.22% 	|
> |   [4]   	| 98.89%  	| 97.87% 	| 97.23% 	| 95.81% 	| 96.71% 	| 95.62% 	| 94.51% 	|
> |   Ours  	| 98.61%  	| 97.75% 	| 97.74%	| 97.75% 	| 96.47% 	| 95.91% 	| 95.41% 	|
>
> Reference:
>
> [1] Towards deep learning models resistant to adversarial attacks, 2019
>
> [2] Theoretically principled trade-off between robustness and accuracy, 2019
>
> [3] Solving a Class of Non-Convex Min-Max Games Using Iterative First Order Methods, 2019
>
> [4] A single-loop smoothed gradient descent-ascent algorithm for nonconvex-concave min-max problems, 2020
>
> [5] Explaining and harnessing adversarial examples, 2015
>
> [6] Adversarial machine learning at scale, 2017

---

### Author Response · Authors · 2021-11-16
**Response to the reviewers**

We thank all reviewers for their time and valuable comments. We are happy that the reviewers find our theoretical results sound (R1, R2, R3), note that the paper is well-written and well-motivated (R1, R2, R3), and mention that the experiments in our work are successful (R1, R2, R3).

We would like to highlight three key points:

**1. Analysis of convex-concave and convex-nonconcave problems:**

Although it is not the focus of our work given the 9 page limit, we tried our best to incorporate suggestions from R1 and R2 to make our paper even stronger. We provided theoretical analysis of GDA-AM for both convex-concave and convex-nonconcave problems in our revision. Please see our reply to R1 and R2 for details.

**2. More discussion on differences between obtained rates and existing works:**

We appreciate R3's suggestion about more discussion of our results. We have added explanations of how to obtain Theorems 4.1 and 4.2. Also, we have added extra discussion of differences between our obtained rates and existing ones, as well as the experimental comparison of GDA-AM and EG+Positive Momentum.

**3. R4:**

We respectfully disagree with your comment regarding the novelty of our work as well as the Crouzeix's bound. We suspect this is due to some misunderstanding. Please see our reply below and Appendix B.1 in our revision for detailed proof.

We hope we have addressed most concerns through our response below and with the revision!

---

### Author Response · Authors · 2021-11-23
**Summary of revision**

## Update: Summary of revision

We sincerely thank your for your time and suggestions. We tried our best to incorporate the suggestions from all the reviewers and highlight the advantages of GDA-AM. Since we can not further update our draft, the summary of the main changes in our revision are shown below.

**a)**  As suggested by R1, we added a discussion about differences between other references in this field in Section 5.

**b)** We provided analysis of our method for convex-concave (B.4), bilinear-quadratic games (B.4.1) and stochastic convex-nonconcave problems (B.5).

**c)** As suggested by R3, we summarized the theoretical advantages of GDA-AM in Appendix B.3. We also added explanation regarding our rates and comparison between other rates in Appendix B.3.

**d)** As suggested by R4, we added the discussion about deriving theoretical results based on the *numerical range* in Section 4 (page 5), B.1 (Bilinear games), and B.4.1 (bilinear-quadratic games).

**e)** We clarified the difficulty and novelty of our theoretical results in Appendix B.1.

From an experimental perspective:

**f)** As suggested by R1, we added numerical comparisons with [1,2,3,4] on Robust Neural Network Training in Appendix C.2.

**g)** As suggested by R3, we added comparisons of our method with EG + positive momentum in Appendix C.1.

Finally, we would like to re-emphasize several **technical contributions** that might have been missed (which can also be found in our response to R2):


**1:** Our obtained Theorem 4.1 and 4.2 provide **nonasymptotic** guarantees, while most other work are asymptotic. Recent work like EG with positive momentum can achieve an **asymptotic** rate of $1-O(1/\sqrt{\kappa})$ under strong assumptions (3 hyperparamteres).

**2:** Our contribution is not just about fix the convergence issue of GDA by applying Anderson Mixing; another contribution is that we arrive at a convergent and tight bound on the **original work** and not just adopting existing analyses. We developed Theorem 4.1 and Theorem 4.2 from a new perspective because applying existing theoretical results (i.e, GMRES and Crouseix's bound) fail to give us neither convergent nor tight bounds.

**3:** The novel Theorem 4.1 only requires **mild conditions** and reflects how the table size $p$ controls the convergence rate. It is also **independent** of the learning rate $\eta$. However, the convergence results of other methods like EG and OG depend on the learning rate, which may yield less than desirable results for ill-specified learning rates.

We hope the updated revision and the following responses can address your concerns and help you better evaluate our work.

Reference:

[1] Towards deep learning models resistant to adversarial attacks, 2019

[2] Theoretically principled trade-off between robustness and accuracy, 2019

[3] Solving a Class of Non-Convex Min-Max Games Using Iterative First Order Methods, 2019

[4] A single-loop smoothed gradient descent-ascent algorithm for nonconvex-concave min-max problems, 2020

---

### Decision · Program_Chairs · 2022-01-20

**Decision:**

Accept (Poster)

**Comment:**

This paper proposes to use Anderson Acceleration on min-max problems, provides some theoretical convergence rates and presents numerical results on toy bilinear problems and GANs.

After the discussion, the reviewers agreed that this paper makes a nice contribution to ICLR. Some concerns were originally expressed in terms of incrementality of the theoretical results with respect to previous work (KCYs, gBHU), but the authors have well clarified their contributions in the discussion, and have updated their manuscript accordingly. There were also initial concerns about the related work coverage, but this was also properly addressed in the rebuttal, with additional experimental comparisons as well as extended related work section, as well as an additional convergence result for convex-nonconcave problems.